# HexFormer: Hyperbolic Vision Transformer with Exponential Map Aggregation

## Abstract

Data across modalities such as images, text, and graphs often contains hierarchical and relational structures, which are challenging to model within Euclidean geometry. Hyperbolic geometry provides a natural framework for representing such structures. Building on this property, this work introduces HexFormer, a hyperbolic vision transformer for image classification that incorporates exponential map aggregation within its attention mechanism. Two designs are explored: a hyperbolic ViT (HexFormer) and a hybrid variant (HexFormer-Hybrid) that combines a hyperbolic encoder with an Euclidean linear classification head. HexFormer incorporates a novel attention mechanism based on exponential map aggregation, which yields more accurate and stable aggregated representations than standard centroid based averaging, showing that simpler approaches retain competitive merit. Experiments across multiple datasets demonstrate consistent performance improvements over Euclidean baselines and prior hyperbolic ViTs, with the hybrid variant achieving the strongest overall results. Additionally, this study provides an analysis of gradient stability in hyperbolic transformers. The results reveal that hyperbolic models exhibit more stable gradients and reduced sensitivity to warmup strategies compared to Euclidean architectures, highlighting their robustness and efficiency in training. Overall, these findings indicate that hyperbolic geometry can enhance vision transformer architectures by improving gradient stability and accuracy. In addition, relatively simple mechanisms such as exponential map aggregation can provide strong practical benefits. Code is available at: Paper under double-blind review

## 1 Introduction

Hyperbolic geometry has emerged as a powerful tool in deep learning due to its ability to model hierarchical and relational structures more effectively than Euclidean spaces (Nickel & Kiela, 2017; Bdeir et al., 2023). Vision Transformers (ViTs) form the backbone of many state-of-the-art vision models (Siméoni et al., 2025; Ravi et al., 2024). However, the reliance on Euclidean representations suggests they may struggle to capture complex, hierarchical structures in ways that hyperbolic spaces naturally can (Chamberlain et al., 2017).

Recent works have attempted to improve the performance by integrating hyperbolic geometry into transformer architectures. Early efforts mapped ViT outputs into hyperbolic space for metric learning (Ermolov et al., 2022), while more recent approaches such as HVT (Fein-Ashley et al., 2024) and LViT from the HyperCore framework (He et al., 2025b) incorporated hyperbolic operations into transformer modules. Although these methods reported improvements, they either restricted hyperbolic integration to limited components or did not investigate training dynamics in detail.

This work introduces a hyperbolic Vision Transformer based on the Lorentz model for image classification. The proposed approach consistently outperforms Euclidean baselines across multiple datasets, activation functions, and model scales, while also surpassing prior hyperbolic ViTs such as HVT (Fein-Ashley et al., 2024) and LViT (He et al., 2025b). Beyond accuracy gains, hyperbolic models also exhibit more stable training dynamics, reducing sensitivity to warmup schedules and hyperparameter tuning.

The main contributions are as follows:

- **Hyperbolic ViT with novel attention (HexFormer)**: A transformer architecture entirely formulated in the Lorentz model of hyperbolic space is developed. In addition, a new attention mechanism incorporating a simple exponential map aggregation is introduced as an alternative to centroid based averaging, providing more stable and effective feature aggregation. This model yields better performance compared to Euclidean ViTs and prior hyperbolic ViT models.
- **Hybrid encoder-classifier design (HexFormer-Hybrid)**: A model that combines a hyperbolic encoder with an Euclidean linear classification head. This further improves performance over both hyperbolic and Euclidean models.
- **Analysis of stable and scalable training**: A detailed study of training dynamics demonstrates that hyperbolic ViTs achieve improved gradient stability and robustness to warmup strategies, reducing the need for extensive fine-tuning.

Extensive experiments demonstrate that both hyperbolic and hybrid models achieve consistent improvements over Euclidean ViTs, with the hybrid variant delivering the strongest results overall. These findings indicate that hyperbolic geometry provides a principled and scalable approach to enhancing transformer architectures in terms of both representational capacity and training stability.

## 2 RELATED WORK

### 2.1 HYPERBOLIC GEOMETRY IN DEEP LEARNING

Hyperbolic embeddings have been employed extensively to encode hierarchical relationships in data (Nickel & Kiela, 2017; Bdeir et al., 2023; Chen et al., 2021). Seminal work projected taxonomies into the Poincaré ball (Nickel & Kiela, 2017). However, previous works have shown that the Poincaré ball suffers from training instabilities (Mishne et al., 2024). In computer vision, convolutional neural networks have been adapted to hyperbolic Lorentz space, yielding performance gains in classification and segmentation (Bdeir et al., 2023). Hyperbolic Lorentz representations have also benefited graph learning and fully connected architectures, particularly when modeling relational data (Yang et al., 2024).

### 2.2 HYPERBOLIC VISION TRANSFORMERS

Vision transformers (Dosovitskiy et al., 2020) have shown state-of-the-art performance for image classification tasks. Nevertheless, they require a lot of data and tend to not perform so well with smaller datasets (Pandey et al., 2023). Due to this, a natural extension is to improve the ViT's architecture to reduce the amount of data needed by changing the representation space. Several approaches have explored integrating hyperbolic geometry into Vision Transformers:

Hyp-ViT projects a vanilla ViT's output into the Poincaré ball and employs a pairwise cross-entropy loss in hyperbolic distance for metric learning, outperforming Euclidean baselines in retrieval tasks (Ermolov et al., 2022).

HVT implements a Vision Transformer on ImageNet with self-attention adapted via Möbius transformations and hyperbolic distance within the Poincaré ball framework (Fein-Ashley et al., 2024).

HyperCore introduces modules for constructing hyperbolic foundation models in both Lorentz and Poincaré geometries. It includes a hyperbolic ViT in Lorentz space (LViT), demonstrating improvements over Euclidean ViTs in image classification (He et al., 2025b). However, these lifts are only observed when finetuning from a bigger dataset, but not when training only on the downstream task.

### 2.3 AGGREGATION AND TRAINING DYNAMICS IN HYPERBOLIC MODELS

Most existing hyperbolic ViT variants use centroid based aggregation in attention mechanisms (Chen et al., 2021; Fein-Ashley et al., 2024; He et al., 2025b). These can introduce distortion in curved spaces when processing big values (Mishne et al., 2024). Aggregation via exponential maps has not been explored in this context, although logarithmic and exponential maps have been used in

hyperbolic models, they have not been applied directly as the aggregation step in attention. Despite the intuitive approach it is an alternative to the expensive fréchet mean. For instance, HVT (Fein-Ashley et al., 2024) maps queries, keys, and values into tangent space and performs the entire attention computation there, which mitigates the benefits from the exponentially growing hyperbolic distance metric. This omission can lead to suboptimal aggregation, as it fails to fully leverage the geometric properties of hyperbolic space. HexFormer approach calculates the scores based on the hyperbolic distance and uses them to aggregate the points on the tangent plane, which allows us leveraging both approaches. Furthermore, the training behavior of hyperbolic transformers (specifically gradient stability and sensitivity to warmup) remains underexamined.

Prior work on hyperbolic models have demonstrated the potential of integrating both Euclidean and non-Euclidean geometry for improving performance (Ermolov et al., 2022; Bdeir et al., 2023). This motivates exploring hybrid approaches that leverage the strengths of both geometries.

## 3 PRELIMINARIES

Hyperbolic geometry is a non-Euclidean geometry with constant negative curvature $K < 0$. Several equivalent models can represent hyperbolic space, such as the Poincaré ball model (Ganea et al., 2018), the Poincaré half-plane model (Tifrea et al., 2018), the Klein model (Gulcehre et al., 2018), and the Lorentz (hyperboloid) model (Nickel & Kiela, 2018). These models are isometric to one another, meaning transformations between them preserve hyperbolic distances (Ramsay & Richtmyer, 1995). The Lorentz model is frequently adopted in hyperbolic deep learning due to its numerical stability and the closed-form expressions of its distance function, exponential map, and logarithmic map (Chen et al., 2021; Yang et al., 2024).

### 3.1 THE LORENTZ MODEL

The $n$-dimensional Lorentz model is the Riemannian manifold $\mathbb{L}_K^n = (\mathcal{L}^n, \mathfrak{g}_x^K)$. It is defined in the $(n + 1)$-dimensional Minkowski space based on the Riemannian metric tensor, using $\mathrm{diag}(-1, 1, \ldots, 1)$. The hyperboloid is then given by

$$\mathcal{L}^n := \{x \in \mathbb{R}^{n+1} \mid \langle x, x \rangle_{\mathcal{L}} = 1/K, \ x_t > 0\},$$

where $K \in \mathbb{R}^-$ is the negative curvature and the Lorentzian inner product is given as

$$\langle x, y \rangle_{\mathcal{L}} := -x_t y_t + x_s^\top y_s = x^\top \mathrm{diag}(-1, 1, \ldots, 1)\, y.$$

which corresponds to the upper sheet of the two-sheeted hyperboloid. The first coordinate $x_t$ is often referred to as the time-like component, while the remaining $n$ coordinates represent spatial dimensions $x_s$, reflecting the connection to the geometry of special relativity. Each point in $\mathbb{L}_K^n$ has the form $x = \begin{bmatrix} x_t \\ x_s \end{bmatrix}, x \in \mathbb{R}^{n+1}, x_t \in \mathbb{R}, x_s \in \mathbb{R}^n$ and $x_t = \sqrt{||x_s||^2 - 1/K}$ (Chen et al., 2021; Bdeir et al., 2023).

### 3.2 TANGENT SPACE

At each point $x \in \mathbb{L}_K^n$, the tangent space is the Euclidean subspace orthogonal to $x$ under the Lorentzian inner product:
$$\mathcal{T}_x \mathbb{L}_K^n := \{y \in \mathbb{R}^{n+1} \mid \langle y, x \rangle_{\mathcal{L}} = 0\}.$$
Although the ambient space is Minkowski, each tangent space is Euclidean. This property allows standard Euclidean operations to be performed in $\mathcal{T}_x \mathbb{L}_K^n$ before mapping results back to the manifold via the exponential map. The tangent space at the origin is denoted as $\mathcal{T}_0 \mathbb{L}_K^n$.(Chen et al., 2021).

### 3.3 EXPONENTIAL AND LOGARITHMIC MAPS

The exponential and logarithmic maps connect the hyperbolic manifold $\mathbb{L}_K^n$ with its tangent spaces $\mathcal{T}_x \mathbb{L}_K^n$. For $x \in \mathbb{L}_K^n$ and $z \in \mathcal{T}_x \mathbb{L}^n$, the exponential map $\exp_x^K(z) : \mathcal{T}_x \mathbb{L}_K^n \to \mathbb{L}_K^n$, maps tangent vectors to hyperbolic spaces:

$$\exp_x^K(z) = \cosh(\alpha)\, x + \sinh(\alpha)\, \frac{z}{\alpha}, \quad \alpha = \sqrt{-K}\, \|z\|_{\mathcal{L}}, \quad \|z\|_{\mathcal{L}} = \sqrt{\langle z, z \rangle_{\mathcal{L}}}.$$

Conversely, for $y \in \mathbb{L}_K^n$, the logarithmic map $\log_x^K(y) : \mathbb{L}_K^n \to \mathcal{T}_x \mathbb{L}_K^n$, os the inverse of exponential map that gives the tangent vector at $x$ pointing toward $y$:

$$\log_x^K(y) = \frac{\cosh^{-1}(\beta)}{\sqrt{\beta^2 - 1}} (y - \beta x), \qquad \beta = K \langle x, y \rangle_{\mathcal{L}}.$$

Where $\log_x^K(\exp_x^K(z)) = z$. These operators provide a consistent mechanism to switch between the curved geometry of $\mathbb{L}_K^n$ and its Euclidean tangent approximations, and are essential for optimization and representation learning in hyperbolic neural networks (Chen et al., 2021; Yang et al., 2024).

## 4 METHODOLOGY

This section describes the Hyperbolic Vision Transformer (HexFormer) variants and their building blocks, and then presents the hyperbolic attention mechanism with a comparison between centroid aggregation and exponential-map aggregation. Implementation choices that follow prior work are explicitly identified and modifications are highlighted. The Lorentz model notation and basic manifold maps follow the preliminaries in Section Preliminaries 3.

### 4.1 MODEL
VARIANTS AND COMPONENT OVERVIEW

Three model variants are considered:

**Euclidean ViT:** a conventional Vision Transformer fully operating in Euclidean space. The main implementation change compared to the original formulation of ViTs (Dosovitskiy et al., 2020) is the adoption of pre-layer normalization instead of post-layer normalization (Radford et al., 2019).

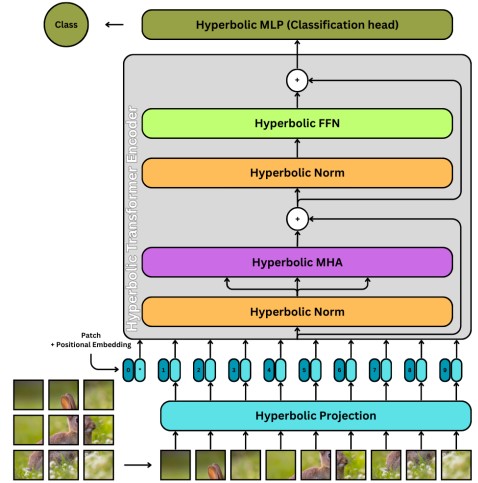

Figure 1: Overview of HexFormer architecture. Hyperbolic embedding, transformer encoder, and classification head operate on the Lorentz manifold. Residual connections are applied to space-like components only.

**HexFormer (hyperbolic ViT):** a transformer whose patch embeddings, encoder layers (attention, MLP), normalization, and classification head are defined on the Lorentz, as visually presented in Figure 1.

**HexFormer-Hybrid:** a hybrid design that uses a Lorentz encoder for feature extraction and an Euclidean linear classifier as the final head.

Below, each major component is described, highlighting which parts are drawn from prior work, how they were combined, and which modifications were introduced to form the final HexFormer design.

**Hyperbolic fully-connected layers (LorentzFC):** The feedforward network (FFN) extends the formulation of HyboNet (Chen et al., 2021), who introduced Lorentz linear layers for hyperbolic neural networks. In their design, the time-like component is computed as

$$y_0 = \lambda \, \sigma(\mathbf{v}^\top \mathbf{x} + b) + \epsilon,$$

where $\sigma$ is a sigmoid, $\lambda$ controls the range, $b$ is a learnable bias, and $\epsilon > \sqrt{1/K}$ ensures numerical validity. The space-like components are then rescaled such that the Lorentz norm constraint

$$\|\mathbf{y}_{1:n}\|^2 - y_0^2 = 1/K$$

is satisfied, placing outputs on the Lorentz manifold.

The LorentzFC implementation in HexFormer builds on the core formulation of Chen et al. (2021) and incorporates later adaptations, including multi-head support and optional normalization from

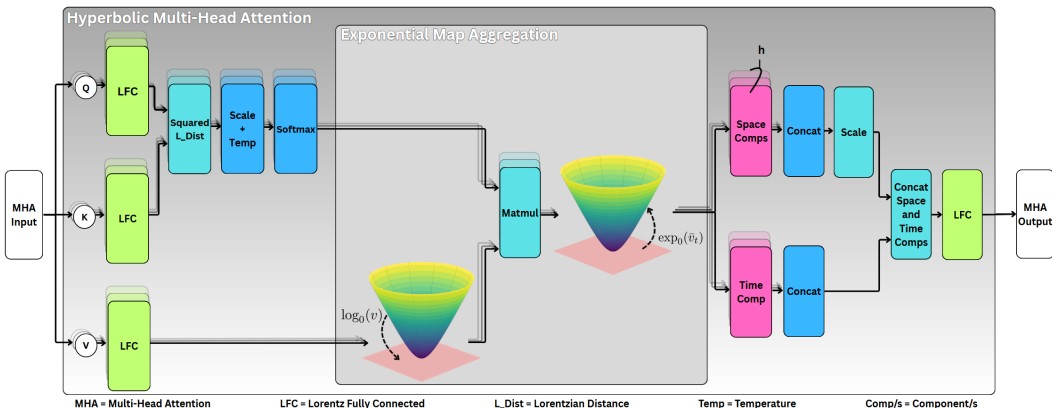

Figure 2: HexFormer multi-head attention. Queries, keys, and values are projected via Lorentz fully connected layers. Scores are computed using Lorentzian distances, aggregated in tangent space via exponential map, and recombined after scaling the space components (to normalize magnitudes and maintain stability in the Lorentz model), then passed as output through a Lorentz FC layer.

HCNN (Bdeir et al., 2023) and curvature-aware time offsets from LResNet (He et al., 2025a). Additional refinements introduced in this work include a unified formulation that integrates multi-head processing, normalization, and curvature-aware offsets into a single flexible module, as well as broadcasting of the time component across heads to ensure Lorentz constraints are satisfied, maintaining Lorentzian geometry while improving stability and efficiency in transformer architectures.

**Patch embeddings and Lorentz batch normalization:** Following the approach of HCNN (Bdeir et al., 2023), image patches are first projected into the ambient Minkowski space and then transformed using a LorentzFC layer. Positional embeddings are added to the space-like components, and a classification token, initialized directly on the manifold, is prepended to the sequence. The time-like component is then computed according to the Lorentz norm constraint (Section 3), producing the complete hyperbolic embedding. lorentz batch normalization, taken from HCNN (Bdeir et al., 2023), is applied in each Transformer block to ensure the normalized features remain consistent with the Lorentz geometry.

**Classification Head:** For classification, the Lorentz MLR formulation of HCNN (Bdeir et al., 2023) is employed, which generalizes multinomial logistic regression to the Lorentz manifold by measuring the hyperbolic distance of embeddings to class-defining hyperplanes. This approach naturally respects the geometry of the hyperbolic space and provides logits suitable for standard cross-entropy training. For HexFormer-Hybrid, an Euclidean linear head (standard classifier) is applied to the flattened Euclidean projection of the CLS token, allowing the model to leverage both hyperbolic and Euclidean representations for classification.

## 4.2 HYPERBOLIC ATTENTION AND AGGREGATION

The attention mechanism is the main point of departure. The attention score computation uses Lorentzian geometry and the squared Lorentzian distance between query and key embeddings; aggregation of values is performed by exponential map aggregation rather than by computing centroids directly on the manifold. The architecture is shown in Figure 2.

**Score computation:** Rather than computing dot products in Euclidean space, the hyperbolic attention mechanism computes scores based on the squared Lorentzian distance (Law et al., 2019), which is defined as:

$$d^2(x, y) = -2k - 2\langle x, y \rangle_{\mathcal{L}}, \tag{1}$$

where $k$ is a curvature-related constant and $\langle \cdot, \cdot \rangle_{\mathcal{L}}$ denotes the Lorentzian inner product. Queries and keys are generated by LorentzFC layers and live on $\mathbb{L}_k^n$. The pairwise compatibility is computed from the Lorentzian inner product $\langle \cdot, \cdot \rangle_{\mathcal{L}}$ and the geodesic distance. With curvature parameter

$K < 0$ the (squared) geodesic distance may be expressed via the Lorentzian product; scores are derived from negative squared distance and scaled. These distances, after appropriate scaling with a scale and a learnable temperature parameter, are passed through the softmax function to obtain the attention weights:

$$\text{score} = \text{softmax}\left(\frac{-d^2(Q, K)}{\sqrt{d_k}} \cdot \tau\right), \tag{2}$$

where $\tau$ is the temperature parameter. This temperature was also added to the Euclidean Transformer to ensure a fair comparison. Typically, the scale is fixed as $\sqrt{d_k}$, but by introducing an additional learnable scale parameter, referred to as temperature, the results improved.

**Exponential aggregation:**   In standard attention mechanisms, the softmax scores are Euclidean scalars that weight the value vectors before aggregation. Directly applying these weights in hyperbolic space is not geometrically consistent. To address this, an exponential aggregation scheme is adopted. First, each values vector $V$ is mapped from the Lorentz manifold $\mathbb{L}_K^n$ to the tangent space at the origin using the logarithmic map,

$$\tilde{V} = \log_0^k(V) \tag{3}$$

The softmax score is then applied as a scalar weight in the tangent space by performing a dot product,

$$u = \sum_i \alpha_i \tilde{V}_i \tag{4}$$

where $\alpha \in \mathbb{R}$ and $u \in \mathcal{T}_0\mathbb{L}_K^n$. Finally, the result is mapped back to the manifold via the exponential map, ensuring that the aggregation remains Lorentz-consistent,

$$h_{agg} = \exp_0^k(u) \tag{5}$$

where $h_{agg} \in \mathbb{L}_k^n$. This procedure preserves the hyperbolic geometry while allowing Euclidean attention weights to modulate the contribution of value vectors in a geometrically valid manner. By operating in the tangent space for scalar multiplications and returning to the manifold through the exponential map, the method avoids distortion, leading to more stable training and better alignment of attention outputs with the manifold's curvature.

## 5   EXPERIMENTS

This section describes the experimental setup, training strategy, and results for evaluating Euclidean, Hyperbolic (HexFormer), and Hybrid (HexFormer-Hybrid) Vision Transformers, along with baseline models HVT (Fein-Ashley et al., 2024) and LViT (He et al., 2025b), on CIFAR-10 (Krizhevsky et al., 2009), CIFAR-100 (Krizhevsky et al., 2009), and Tiny-ImageNet (Le & Yang, 2015). The experiments assess the effectiveness of hyperbolic representations in improving performance, training stability, and optimization dynamics. In addition, results are reported across different ViT scales (Tiny, Small, and Base), and further experiments analyze the effect of centroid versus ExpAgg aggregation strategies, as well as gradient stability under warmup and no warmup schedules.

### 5.1   TRAINING STRATEGY

All models were trained under a unified training pipeline, differing only in optimizer choice, learning rate, and weight decay to respect the underlying geometry. Standard data augmentations, weight initialization, loss functions, and learning rate scheduling were applied consistently across experiments. The full training strategy, together with hyperparameter settings and implementation details, are provided in Appendix A.1.

### 5.2   COMPARISON OF HYPERBOLIC VISION TRANSFORMERS

Table 1 presents the accuracy comparison of various Vision Transformer (ViT) based models on CIFAR-10, CIFAR-100, and Tiny-ImageNet. HexFormer consistently outperforms the Euclidean ViT, as well as HVT (Fein-Ashley et al., 2024) and LViT (He et al., 2025b), across all datasets. The HexFormer-Hybrid variant achieves the highest accuracy overall, surpassing both HexFormer

Table 1: Accuracy comparison on CIFAR-10, CIFAR-100, and Tiny-ImageNet. The reported hyperbolicity values ($\delta_{rel}$) are taken from HCNN (Bdeir et al., 2023) and are normalized by the dataset diameter.

| Model | CIFAR-10 ($\delta_{rel} = 0.26$) | CIFAR-100 ($\delta_{rel} = 0.23$) | Tiny-ImageNet ($\delta_{rel} = 0.20$) |
|---|---|---|---|
| Euclidean ViT | $93.38 \pm 0.118$ | $74.64 \pm 0.408$ | $60.74 \pm 0.356$ |
| HVT[†] (Fein-Ashley et al., 2024) | 61.44 | 42.77 | 40.12 |
| LViT[†] (He et al., 2025b) | 85.02 | 69.11 | 53.01 |
| HexFormer (ours) | $\underline{93.42} \pm 0.183$ | $\underline{75.65} \pm 0.434$ | $\underline{60.87} \pm 0.635$ |
| HexFormer-Hybrid (ours) | $\mathbf{93.53} \pm 0.287$ | $\mathbf{75.85} \pm 0.337$ | $\mathbf{62.93} \pm 0.447$ |

[†]: Results for HVT and LViT are taken from the LViT paper (He et al., 2025b).

and the Euclidean baseline. Notably, the Euclidean ViT, HexFormer, and HexFormer-Hybrid results reported in Table 1 are based on the ViT-Tiny architecture, whereas the results for HVT and LViT were reported on the larger ViT-Base model. Despite operating with significantly fewer parameters, HexFormer-Tiny and HexFormer-Hybrid-Tiny achieve higher accuracy, demonstrating the efficiency and representational power of hyperbolic models. Details on the ViT architecture variants (Tiny, Small, and Base) are provided in Table 5 in Appendix A.1.

The Euclidean results correspond to a ViT implementation trained from scratch under the same conditions as HexFormer, ensuring a fair comparison, for more details on the implementation refer to Appendix A.1. The results reported for HVT (Fein-Ashley et al., 2024) and LViT (He et al., 2025b) are taken directly from the LViT paper (He et al., 2025b); while LViT reports higher accuracy on these datasets when fine-tuned after pretraining on ImageNet (Deng et al., 2009b), the values shown here correspond to the models trained directly on each dataset without pretraining, as reported in their table. Even under these comparable training conditions, HexFormer surpasses the Euclidean baseline by a larger margin than that observed for LViT in their pre-trained results. The consistent improvement over the Euclidean baseline highlights the effectiveness of HexFormer and its hybrid variant compared to other hyperbolic baselines.

### 5.3 WARMUP DEPENDENCY AND GRADIENT STABILITY

During preliminary experiments, it was observed that in the early stages of training (first few epochs with warmup), HexFormer and HexFormer-Hybrid consistently outperformed their Euclidean counterparts before performance converged. This behavior is illustrated in Figure 4 in Appendix A.2.3. Furthermore, Appendix A.2.4 (Table 11) reports experiments where models were trained for only a limited number of epochs without warmup. The results indicate that HexFormer and HexFormer-Hybrid achieve substantially higher accuracy than Euclidean ViT in short training regimes, suggesting that hyperbolic representations accelerate the initial learning phase. To further investigate this accelerated early training and its relation to warmup dependency, experiments were conducted with and without warmup across datasets and architectures. As reported in Table 2, HexFormer and HexFormer-Hybrid show minimal performance differences between warmup and no-warmup conditions, while Euclidean ViTs exhibit a more pronounced dependency on warmup, with larger performance gaps when tested on CIFAR-10 and Tiny-ImageNet. This trend is consistent also on CIFAR-100 and across different ViT scales (Tiny, Small, and Base), as detailed in Table 3.

These results suggest that hyperbolic models exhibit more stable gradients, reducing sensitivity to warmup scheduling. To confirm this, absolute gradient histograms were plotted in the same manner as in RAdam (Liu et al., 2019), where gradient instability was used to motivate the necessity of warmup. As shown in Figure 3, gradients in HexFormer remain more stable than those in Euclidean ViTs when tested without warmup, where Euclidean ViT models exhibit higher distortion in their gradient distributions, especially during the early updates. HexFormer-Hybrid demonstrates similar stability, with additional comparisons provided in Appendix A.2.3. This stability may arise from the geometric properties of hyperbolic space. The negative curvature tends to spread representations and constrain the scale of activations, helping to prevent extreme values in the forward pass and,

Table 2: Accuracy comparison between HexFormer, HexFormer-Hybrid and Euclidean ViT on CIFAR-10 and Tiny-ImageNet under warmup and no warmup. CIFAR-10 and Tiny-ImageNet are averaged over 3 seeds.

| Model | CIFAR-10 | | Tiny-ImageNet | |
|---|---|---|---|---|
| | Warmup = 0 | Warmup = 10 | Warmup = 0 | Warmup = 10 |
| Euclidean ViT | $91.75 \pm 0.249$ | $93.38 \pm 0.118$ | $59.46 \pm 0.381$ | $60.74 \pm 0.356$ |
| HexFormer | $\underline{93.55} \pm 0.424$ | $\underline{93.42} \pm 0.183$ | $\underline{60.12} \pm 0.275$ | $\underline{60.87} \pm 0.635$ |
| HexFormer-Hybrid | $\mathbf{93.91} \pm 0.211$ | $\mathbf{93.53} \pm 0.287$ | $\mathbf{62.17} \pm 0.295$ | $\mathbf{62.93} \pm 0.447$ |

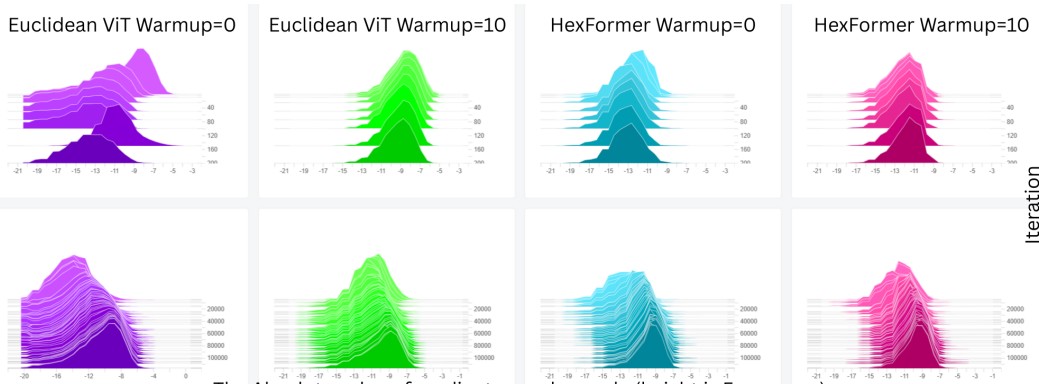

Figure 3: The histogram of the absolute value of gradients on a log scale during the training of ViT-Small on CIFAR-100 comparing HexFormer vs. Euclidean ViT with and without warmup (offset mode). Top row shows close-up for first 200 iterations.

consequently, extreme gradients during backpropagation. As a result, hyperbolic models maintain smoother and more consistent gradient distributions throughout training, independent of the choice of optimizer or aggregation method.

### 5.4 VISION TRANSFORMER VARIANTS

Table 3 shows CIFAR-100 performance on different ViT variants. Hexformer and Hexformer-Hybrid models outperform Euclidean models, with larger models amplifying improvements. Table 5 in Appendix A.1 lists ViT variant architectures and parameter counts. Only the ViT-Tiny variants for HexFormer and Euclidean ViT were specifically tuned due to limited computational resources; the hyperparameter choices for all models are provided in Appendix A.1. Although Small and Base ViTs are expected to outperform Tiny-ViT by a larger margin, the limited tuning results in only modest improvements. Nevertheless, the table demonstrates that HexFormer and HexFormer-Hybrid consistently outperform Euclidean models across all scales, indicating that the benefits of hyperbolic representations persist even without extensive tuning. This suggests that hyperbolic feature representations provide robust gains that generalize across model sizes.

### 5.5 AGGREGATION STRATEGIES

Table 4 compares the performance of HexFormer and HexFormer-Hybrid on CIFAR-100 using the centroid aggregation versus the ExpAgg (exponential map aggregation) method under warmup and no warmup (with both methods tuned equally to ensure a fair comparison). While the accuracy differences between centroid and ExpAgg are relatively small, ExpAgg consistently achieves higher performance for both HexFormer and HexFormer-Hybrid. Moreover, ExpAgg provides greater numerical stability. In some cases, centroid can produce NaN/inf values during training, causing the model to stop learning, particularly when used with the AdamW (Loshchilov, 2017) optimizer. Ap-

Table 3: Accuracy for different ViT models scales for HexFormer, HexFormer-Hybrid and Euclidean ViT on CIFAR-100 under warmup and no warmup. Averaged over 10 seeds for ViT-Tiny and 5 seeds for ViT-Small and ViT-Base.

| ViT Variant | Model | CIFAR-100 | |
|---|---|---|---|
| | | Warmup = 0 | Warmup = 10 |
| ViT-Tiny | Euclidean ViT | $71.86 \pm 0.255$ | $74.64 \pm 0.408$ |
| | HexFormer | $\underline{75.06} \pm 0.515$ | $\underline{75.65} \pm 0.434$ |
| | HexFormer-Hybrid | $\mathbf{75.44} \pm 0.337$ | $\mathbf{75.85} \pm 0.337$ |
| ViT-Small | Euclidean ViT | $70.53 \pm 0.451$ | $73.6 \pm 1.046$ |
| | HexFormer | $\underline{75.59} \pm 0.47$ | $\underline{75.54} \pm 0.703$ |
| | HexFormer-Hybrid | $\mathbf{75.74} \pm 0.759$ | $\mathbf{76.09} \pm 0.68$ |
| ViT-Base | Euclidean ViT | $70.72 \pm 0.59$ | $70.54 \pm 0.375$ |
| | HexFormer | $\underline{75.73} \pm 1.074$ | $\underline{76.49} \pm 1.214$ |
| | HexFormer-Hybrid | $\mathbf{76.44} \pm 1.423$ | $\mathbf{76.81} \pm 0.51$ |

Table 4: Accuracy comparison on CIFAR-100 under warmup and no warmup comparing centroid vs ExpAgg. Averaged over 10 seeds.

| Model | CIFAR-100 | |
|---|---|---|
| | Warmup = 0 | Warmup = 10 |
| HexFormer (Centroid) | $74.63 \pm 0.589$ | $75.38 \pm 0.351$ |
| HexFormer (ExpAgg) | $\mathbf{75.06} \pm 0.515$ | $\mathbf{75.65} \pm 0.434$ |
| HexFormer-Hybrid (Centroid) | $75.31 \pm 0.329$ | $75.73 \pm 0.357$ |
| HexFormer-Hybrid (ExpAgg) | $\mathbf{75.44} \pm 0.623$ | $\mathbf{75.85} \pm 0.337$ |

pendix A.3 shows an example figure and explains why HexFormer with centroid often produces NaNs/inf. Overall, ExpAgg not only yields better accuracy but also ensures more reliable and robust training in practice.

Additional experiments evaluating HexFormer and HexFormer-Hybrid under variations in activation functions, optimizers, and training epochs are provided in Appendix A.2. These results demonstrate the robustness of hyperbolic representations, the consistent gradient stability across different settings, and the benefits of the hyperbolic architecture over Euclidean baselines.

## 6    CONCLUSION

HexFormer introduces Lorentzian hyperbolic attention with exponential map aggregation, along with a hybrid design that combines hyperbolic encoder and Euclidean classification. Across CIFAR-10, CIFAR-100, and Tiny-ImageNet, both variants consistently outperform Euclidean ViTs, with the hybrid model delivering the best overall performance. Exponential aggregation proves to be more stable than centroid-based methods, avoiding divergence during training. More broadly, hyperbolic models exhibit smoother gradient dynamics than Euclidean transformers, reducing sensitivity to warmup schedules and enabling faster, more reliable convergence. In addition to surpassing Euclidean baselines, HexFormer also achieves higher accuracy than prior hyperbolic vision transformers, establishing new state-of-the-art results in this setting. These findings show that hyperbolic geometry enhances both representation quality and optimization stability, establishing HexFormer and HexFormer-Hybrid as practical and scalable alternatives for vision tasks.

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

# A APPENDIX

## A.1 TRAINING STRATEGY AND EXPERIMENTS SET-UP

**Data Augmentation.** Random horizontal flips, random cropping with padding, CIFAR10Policy (Cubuk et al., 2018), normalization, and random erasing (probability 0.25) were applied. Dynamic augmentation using CutMix (Yun et al., 2019) or MixUp (Zhang, 2017) with probability 0.5 was applied on-the-fly during training.

**Weight Initialization and Loss:** Xavier Normal initialization (Glorot & Bengio, 2010) and Label Smoothing Cross-Entropy (Szegedy et al., 2016) were used.

**Learning Rate Scheduler:** Cosine Annealing Warmup Restarts (Loshchilov & Hutter, 2016) managed the learning rate, incorporating warmup periods followed by periodic annealing.

**Optimizer:** In the reported experiments, AdamW (Loshchilov, 2017) was used for Euclidean models, and Riemannian AdamW (Bdeir et al., 2024) was used for Hexformer and Hexformer-Hybrid models. In Appendix Subsection A.2 there is an additional experiment to compare both optimizers for both Euclidean ViT and Hexformer in Table 9.

**Hyperparameters and Tuning:** Table 6 lists the training hyperparameters for Tiny models, while Table 7 reports the learning rate and weight decay for the other models, with all remaining hyperparameters matching those in Table 6. Learning rate and weight decay were tuned for Euclidean ViT, HexFormer with ExpAgg and HexFormer with centroid. For the HexFormer-Hybrid the same hyperparameters that were tuned for HexFormer were used without tuning. The tuning was done on CIFAR-100 and the best hyperparameters were applied for CIFAR-10. For Tiny-Imagenet the learning rate and weight decay were tuned. Small-ViT models used scaled learning rates according to:

$$\eta_{new} = \eta \times \sqrt{\frac{\text{hidden dimensions}}{\text{new hidden dimensions}}}.$$

without tuning, and for Base-ViT the equation above gave worse results than expected, so the same learning rate as Small-ViT was applied and it gave better results, also the weight decay was increased.

Table 5: Vision Transformer (ViT) model variants with architectural specifications.

| Model | Patch Size | Hidden Dim | MLP Dim | #Layers | #Heads | Parameters |
|---|---|---|---|---|---|---|
| ViT-Tiny | 4 | 192 | 384 | 9 | 12 | ∼3M |
| ViT-Small | 4 | 384 | 768 | 12 | 6 | ∼14M |
| ViT-Base | 4 | 768 | 3072 | 12 | 12 | ∼85M |

Table 6: Training hyperparameters for Euclidean ViT, HexFormer, and HexFormer-Hybrid using Tiny-ViT size on CIFAR-100 and CIFAR-10.

| Hyperparameter | Euclidean ViT | HexFormer | HexFormer-Hybrid |
|---|---|---|---|
| Number of epochs | 100 | 100 | 100 |
| Warmup | 0 or 10 | 0 or 10 | 0 or 10 |
| Patch size | 4 | 4 | 4 |
| Batch size (train) | 128 | 128 | 128 |
| Batch size (test) | 512 | 512 | 512 |
| Learning rate | $4.15 \times 10^{-3}$ | $4.35 \times 10^{-3}$ | $4.35 \times 10^{-3}$ |
| Weight decay | $6 \times 10^{-2}$ | $5 \times 10^{-2}$ | $5 \times 10^{-2}$ |
| Optimizer | AdamW | Riemannian AdamW | Riemannian AdamW |
| Encoder | Euclidean | Lorentz | Lorentz |
| Classification head | Euclidean MLR | Lorentz MLR | Euclidean MLR |

Table 7: Learning rate and weight decay hyperparameters for Euclidean ViT and HexFormer (the Hyperparameters for HexFormer were also used for HexFormer-Hybrid).

| Hyperparameter | Learning rate | Weight decay |
|---|---|---|
| Tiny-ViT Hexformer with Centroid | $4.65 \times 10^{-2}$ | $5 \times 10^{-2}$ |
| Small-ViT Euclidean | $2.934 \times 10^{-3}$ | $6 \times 10^{-2}$ |
| Small-ViT Hexformer | $3.076 \times 10^{-3}$ | $5 \times 10^{-2}$ |
| Base-ViT Euclidean | $2.934 \times 10^{-3}$ | $1 \times 10^{-1}$ |
| Base-ViT Hexformer | $3.076 \times 10^{-3}$ | $1 \times 10^{-1}$ |
| Tiny-ImageNet dataset | $3 \times 10^{-3}$ | $5 \times 10^{-1}$ |

**Curvature Selection:** All experiments in this work use a fixed curvature of $-1$. Additionally preliminary tests were conducted with alternative fixed curvatures of $-0.5$ and $-2$ for the Tiny-ViT configuration. While training and inference remained stable across these settings, the overall performance was consistently lower, and therefore these variants were not extended to larger models. We also experimented with making the curvature learnable and observed a slight improvement on CIFAR-100 for both warmup and no-warmup training regimes in the Tiny-ViT setting, without introducing any stability issues. These observations suggest that curvature learning is a promising direction for further improving hyperbolic Vision Transformers.

## A.2 ADDITIONAL EXPERIMENTS

This section presents supplementary experiments evaluating HexFormer and HexFormer-Hybrid under various settings, including different activation functions, optimizers, and gradient behavior. All datasets used in the experiments exhibit hierarchical class relations and high hyperbolicity (low $\delta_{\text{rel}}$). In addition to the main HexFormer and hybrid design, an alternative hybrid variant that uses a Euclidean encoder combined with a hyperbolic classification head (MLR) was experimented as well; however, this configuration did not yield competitive performance. The following subsections report results on activation functions, optimizer choices, gradient stability, and epoch-wise performance, highlighting the consistent advantages of hyperbolic representations across settings with hierarchical datasets structures.

**Hyperbolicity:** Hyperbolicity, often defined in the sense of Gromov, measures how close a metric space is to behaving like a tree or a negatively curved space. A space is said to be $\delta$-hyperbolic if all of its geodesic triangles are "$\delta$-slim," meaning each side lies within a small neighborhood of the other two. Equivalently, hyperbolicity can be characterized by a four-point condition based on pairwise distances. Low hyperbolicity values indicate strong tree-like structure, while higher values suggest the space is more Euclidean-like. This property is commonly used to assess whether data or graphs are well suited to hyperbolic representation.

**Choice of Datasets:** Hyperbolic geometry is especially effective for data with hierarchical or coarse-to-fine structure, as distances in negatively curved spaces grow exponentially with radius (Nickel & Kiela, 2017; Ganea et al., 2018). HCNN Bdeir et al. (2023), has shown that datasets like CIFAR-10, CIFAR-100, and Tiny-ImageNet exhibit measurable hierarchical structure, reflected in their hyperbolicity scores (reported in Table 1). Concretely, CIFAR-10 contains latent semantic groupings (animals vs. vehicles), while CIFAR-100 has an explicit 20-superclass hierarchy defined in the dataset annotations. Tiny-ImageNet, being a subset of ImageNet, inherits the well-established WordNet taxonomy on which ImageNet is built; categories follow paths such as mammal →carnivore →canine →dog →husky, providing deep hierarchical relationships Deng et al. (2009a). These properties make all three datasets suitable for evaluating hyperbolic representations.

In addition to these hierarchically structured datasets, Hexformer and Hexformer-Hybrid were evaluated on datasets that would be considered to exhibit weak or flat semantic structure, such as MNIST, Fashion-MNIST, and SVHN. The results are reported in Section A.2.2.

### A.2.1 Additional Experiments on different activation function and on different optimizer

Additional experiments were conducted to evaluate the robustness of HexFormer and HexFormer-Hybrid under variations in activation functions and optimizer choices. Using Leaky ReLU instead of GELU in the encoder feed-forward networks, both HexFormer and HexFormer-Hybrid outperform the Euclidean ViT under warmup and no warmup conditions (Table 8). The gradient stability observed during training is maintained across warmup settings, demonstrating the robustness of the hyperbolic representations to different activation functions.

Table 8: Accuracy comparison between HexFormer and Euclidean ViT on CIFAR-100 with Leaky ReLU activation under warmup and no warmup. Averaged over 5 seeds.

| Model | CIFAR-100 | |
| --- | --- | --- |
| | Warmup = 0 | Warmup = 10 |
| Euclidean ViT | $72.17 \pm 0.43$ | $74.2 \pm 0.24$ |
| HexFormer | $\underline{74.65} \pm 0.407$ | $\underline{74.52} \pm 0.331$ |
| HexFormer-Hybrid | $\mathbf{74.85} \pm 0.459$ | $\mathbf{74.73} \pm 0.378$ |

Experiments with different optimizers further illustrate the behavior of the models. Euclidean ViTs perform better with AdamW (Loshchilov, 2017) (as expected, since Euclidean models do not have manifold parameters and thus do not require Riemannian AdamW (Bdeir et al., 2024)), while hyperbolic models benefit more from Riemannian AdamW (Bdeir et al., 2024) (Table 9). Importantly, the gradient stability is also evident when using standard AdamW (Loshchilov, 2017) for hyperbolic models, indicating that this stability arises from the hyperbolic geometry rather than the optimizer choice.

Table 9: Accuracy comparison between HexFormer and Euclidean ViT on CIFAR-100 under warmup and no warmup, using AdamW (Loshchilov, 2017) and Riemannian AdamW (Bdeir et al., 2024) optimizers.

| Optimizer / Model | AdamW | | RiemannianAdamW | |
| --- | --- | --- | --- | --- |
| | Warmup = 0 | Warmup = 10 | Warmup = 0 | Warmup = 10 |
| Euclidean ViT | $71.86 \pm 0.255$ | $\mathbf{74.64} \pm 0.408$ | $71.49 \pm 0.467$ | $73.7 \pm 0.443$ |
| HexFormer | $\mathbf{74.68} \pm 0.52$ | $74.63 \pm 0.427$ | $\mathbf{75.06} \pm 0.515$ | $\mathbf{75.65} \pm 0.434$ |

### A.2.2 Experiments on weak/flat Hierarchical Datasets:

To further examine the behavior of Hexformer beyond hierarchically structured data, Euclidean ViT, HexFormer, and HexFormer-Hybrid were evaluated on datasets that exhibit minimal or weak hierarchical structure, namely MNIST, Fashion-MNIST, and SVHN. These datasets contain labels corresponding to digits or simple clothing categories and are considered "flat" in terms of semantic hierarchy. Across all three datasets, the results were very similar among the three models. The experiments were ran 3 times on different seeds with warmup 10, the Euclidean ViT achieved slightly higher accuracy, HexFormer slightly outperforms the Hybrid variant on Fashion-MNIST, while the two are essentially tied on MNIST and SVHN. Overall, these findings indicate that hyperbolic architectures do not provide significant advantages when the underlying data lack hierarchical structure, and all three models behave comparably in such settings.

### A.2.3 Gradients Stability

Figures 5 to 9 present histograms of the absolute gradient values in both offset and overlay view modes, comparing HexFormer, HexFormer-Hybrid, and Euclidean ViTs. HexFormer and HexFormer-Hybrid display more stable gradients, maintaining smooth distributions even without warmup, whereas Euclidean models show clear distortions under the same conditions.

Table 10: Results on weak/flat Hierarchical Datasets

| Model | MNIST | Fashion-MNIST | SVHN |
|---|---|---|---|
| Euclidean ViT | **99.41** $\pm$ 0.02 | **94.56** $\pm$ 0.10 | **97.31** $\pm$ 0.04 |
| HexFormer | 99.35 $\pm$ 0.06 | 94.02 $\pm$ 0.20 | 97.13 $\pm$ 0.181 |
| HexFormer-Hybrid | 99.39 $\pm$ 0.01 | 93.86 $\pm$ 0.13 | 97.13 $\pm$ 0.104 |

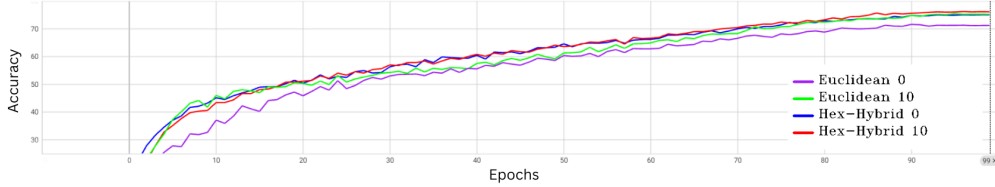

(a) HexFormer-Hybrid vs. Euclidean on CIFAR-100

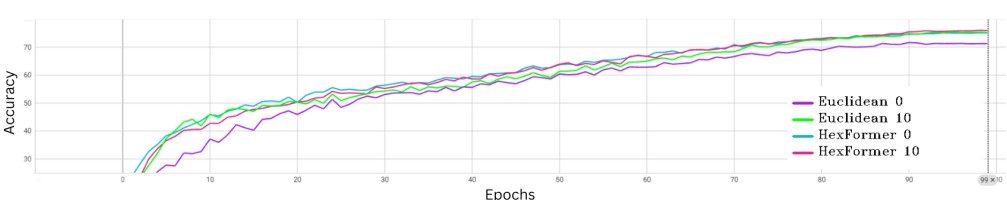

(b) HexFormer vs. Euclidean on CIFAR-100

Figure 4: Comparison of accuracy over epochs on CIFAR-100. The numbers 0 and 10 in the legend refer to the warmup.

### A.2.4 EPOCH-WISE PERFORMANCE ANALYSIS

Table 11 shows accuracy evolution across different epochs on CIFAR-100 (no warmup). HexFormer and HexFormer-Hybrid models consistently outperform Euclidean models at each epoch, highlighting the faster convergence and stronger representational power of hyperbolic architectures.

The HexFormer-Hybrid variant achieves the best performance across all experiments. Hyperbolic encoders capture hierarchical and structured representations effectively, but fully hyperbolic classifiers can be more prone to overfitting (Bdeir et al., 2024; Gao et al., 2022). By using a Euclidean head, the Hybrid model can leverage the benefits of hyperbolic embeddings while reducing the risk of overfitting during the final classification step, resulting in improved generalization and overall performance.

### A.3 CENTROID VS EXPONENTIAL MAP AGGREGATION

As shown by Mishne et al. (2024), computations in Lorentz space using floating point representation may result in numerical instability. For example, for Float32 values that differ in more than 16 order of magnitude can not be added together since there is no space in the mantisa of the bigger value to store the smaller value. This could lead to invalid data points, i.e., there could be data points that do not satisfy the constrain $\langle x, x \rangle_{\mathcal{L}} = 1/K$. This becomes specially problematic whenever the addition is done with squared values as it happens with the Lorentz norm.

All of this results in a training process where models that use centroid might get NaN/inf values, which ends up collapsing the training run. However, when centroid is changed to the Exponential Map aggregation, this collapse is no longer observed as shown in Figure 10. In order explain why this happens, a small example will be done for the Lorentz space $\mathcal{L}^1 := \{x \in \mathbb{R}^2 \mid \langle x, x \rangle_{\mathcal{L}} = 1/K, \ x_t > 0\}$ with $K = -1$.

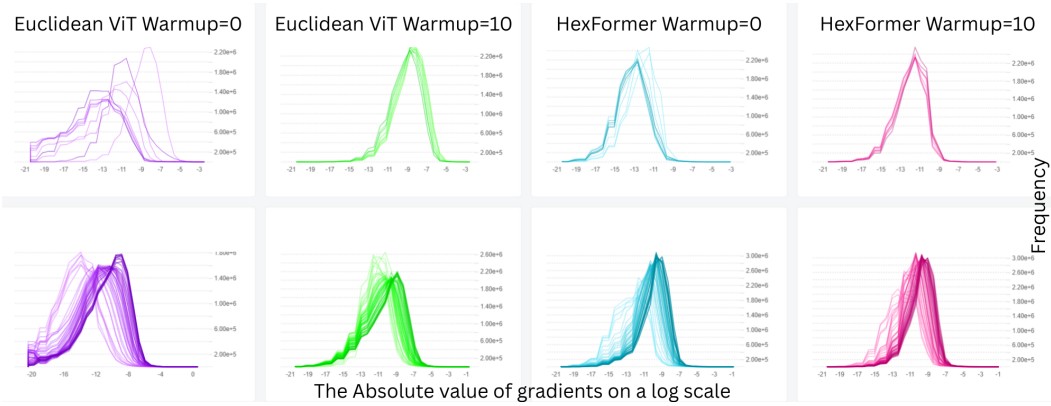

Figure 5: The histogram of the absolute value of gradients on a log scale during the training of ViT-Small on CIFAR-100 comparing HexFormer vs. Euclidean ViT with and without warmup (overlay mode). Top row shows close-up for first 200 iterations.

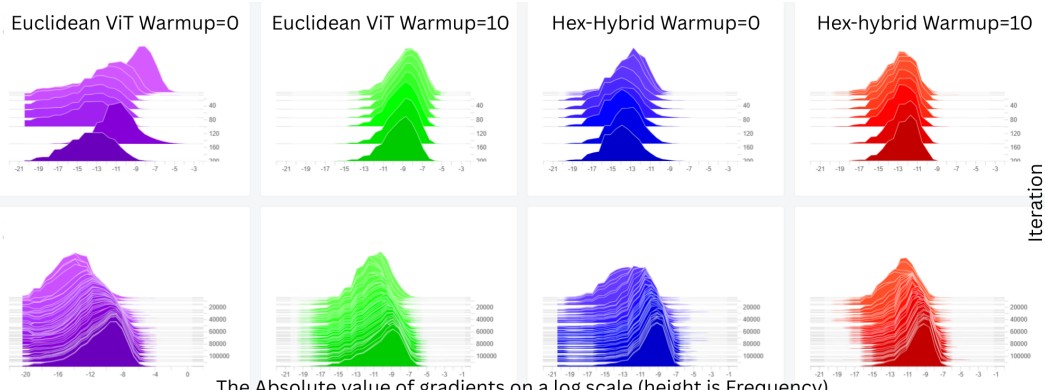

Figure 6: The histogram of the absolute value of gradients on a log scale during the training of ViT-Small on CIFAR-100 comparing HexFormer-Hybrid vs. Euclidean ViT with and without warmup (offset mode). Top row shows close-up for first 200 iterations.

Let's consider the space component from 2 tensors obtained from a *Lorentz linear layer* with values $10^8$ for the tensor x and 0 for the tensor y. Then, in order to calculate the time components the Lorentz constrain has to be solved as follows:

$$x_t : \langle x, x \rangle_{\mathcal{L}} = 1/K \Leftrightarrow -x_t^2 + x_s^2 = 1/K \Leftrightarrow x_t = \sqrt{x_s^2 - 1/K} = \sqrt{10^{16} + 1} \approx 10^8 \tag{6}$$
$$y_t : \langle y, y \rangle_{\mathcal{L}} = 1/K \Leftrightarrow -y_t^2 + y_s^2 = 1/K \Leftrightarrow y_t = \sqrt{y_s^2 - 1/K} = \sqrt{0 + 1} = 1$$

Notice that due to the previously mentioned imprecision in float representation, $x_t$ is approximated to $10^8$ instead of taking the value $\sqrt{10^{16} + 1}$. Let's suppose that the values obtained for the attention scores ($\alpha_1$ and $\alpha_2$) are 0.5 for both x and y. Then the centroid calculation would be:

$$\mu = \frac{V}{\sqrt{-1/K}|||V||_{\mathcal{L}}|} = \frac{\alpha_1 x + \alpha_2 y}{\sqrt{-1/K}|||\alpha_1 x + \alpha_2 y||_{\mathcal{L}}|} \tag{7}$$

Let's calculate first $V$ and $||V||_{\mathcal{L}}$:

$$V = \alpha_1 x + \alpha_2 y = 0.5 \begin{bmatrix} 10^8 \\ 10^8 \end{bmatrix} + 0.5 \begin{bmatrix} 1 \\ 0 \end{bmatrix} = \begin{bmatrix} 5 \cdot 10^7 + 0.5 \\ 5 \cdot 10^7 \end{bmatrix} \approx \begin{bmatrix} 5 \cdot 10^7 \\ 5 \cdot 10^7 \end{bmatrix} \tag{8}$$
$$||V||_{\mathcal{L}} = -V_t^2 + V_s^2 = -(5 \cdot 10^7)^2 + (5 \cdot 10^7)^2 = 0$$

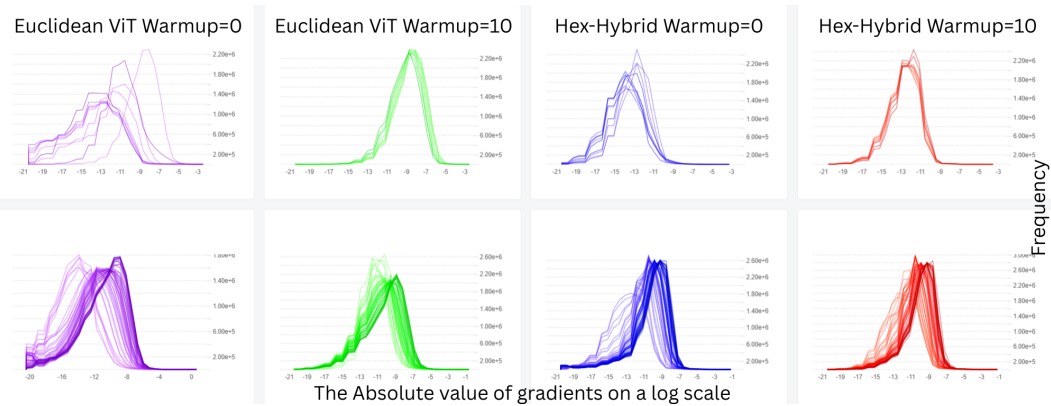

Figure 7: The histogram of the absolute value of gradients on a log scale during the training of ViT-Small on CIFAR-100 comparing HexFormer-Hybrid vs. Euclidean ViT with and without warmup (overlay mode). Top row shows close-up for first 200 iterations.

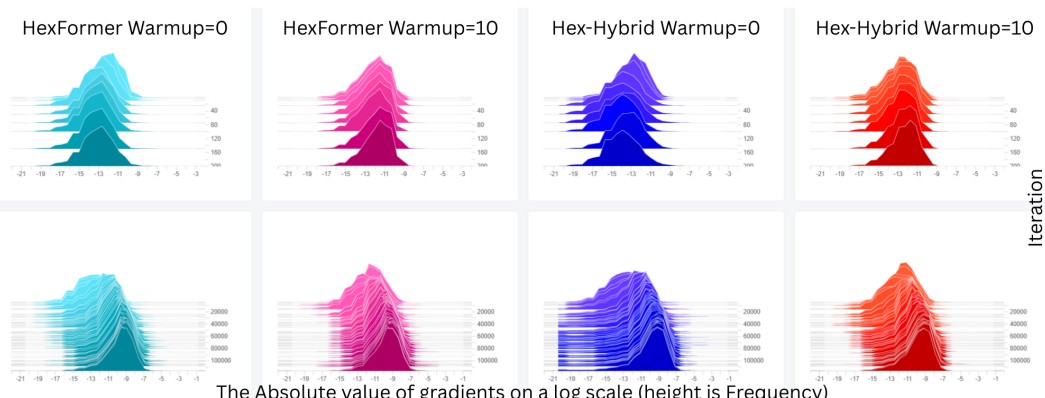

Figure 8: The histogram of the absolute value of gradients on a log scale during the training of ViT-Small on CIFAR-100 comparing HexFormer vs. HexFormer-Hybrid with and without warmup (offset mode). Top row shows close-up for first 200 iterations.

Notice that again due to the representation imprecision, $V_t$ is wrongly calculated which leads to $||V||_{\mathcal{L}} = 0$. This results in a division with a denominator of 0 where NaN/inf values arise:

$$\mu = \frac{V}{\sqrt{-1/K}|||V||_{\mathcal{L}}|} = \frac{V}{0} = NaN/inf \tag{9}$$

This issue can be side stepped using an $\epsilon$ threshold. That way the division can still be done. Since the $\epsilon$ threshold needs to be small to not perturbed the cases where there is no imprecision/error in the calculation, a small value needs to be used. A standard value used is $\epsilon = 10^{-8}$. However, this leads to the backward pass receiving updates with the order of $10^7/10^{-8} = 10^{15}$ which can easily escalate and lead to model divergence as shown in Figure 10.

### A.3.1 FAILED EXPERIMENTS WITH CENTROID

During experiments using the AdamW (Loshchilov, 2017) optimizer, it was observed that Hex-Former with centroid-based aggregation often produced NaN/inf values during training. The figure below compares the same training runs over 5 different seeds. For HexFormer with centroid (Figure 10a), 4 out of 5 seeds encountered NaNs, causing training to fail. In contrast, HexFormer with exponential map aggregation (ExpAgg, Figure 10b) successfully completed training for all 5 seeds, demonstrating the improved numerical stability of the ExpAgg method.

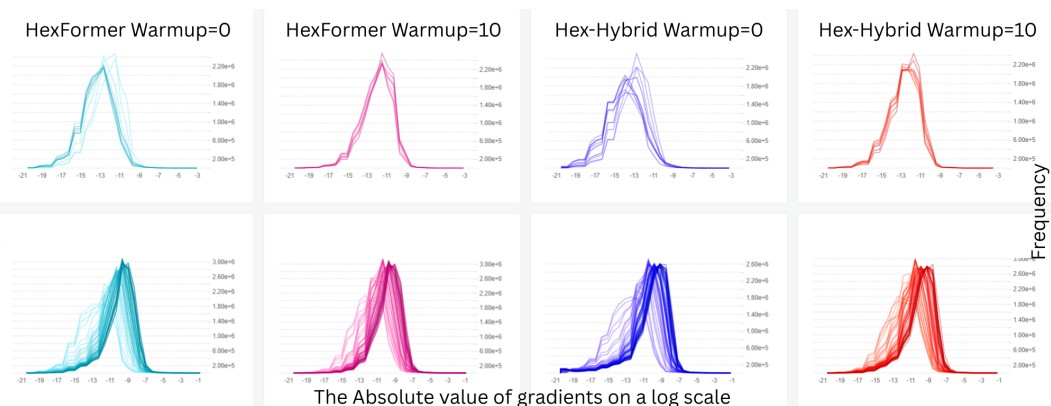

Figure 9: The histogram of the absolute value of gradients on a log scale during the training of ViT-Small on CIFAR-100 comparing HexFormer vs. HexFormer-Hybrid with and without warmup (overlay mode). Top row shows close-up for first 200 iterations.

Table 11: Model accuracy when trained for different epochs between HexFormer, HexFormer-Hybrid and Euclidean ViT on CIFAR-100. Notice that due to the learning rate scheduler, this is not analogous to training for 100 epochs and checking the performance at an intermediate epoch.

| Model | Epochs | CIFAR-100 |
|---|---|---|
| Euclidean ViT | 20 | 55.47 |
| HexFormer | 20 | **58.66** |
| HexFormer-Hybrid | 20 | 58.57 |
| Euclidean ViT | 30 | 62.99 |
| HexFormer | 30 | **65.34** |
| HexFormer-Hybrid | 30 | 64.12 |
| Euclidean ViT | 40 | 66.99 |
| HexFormer | 40 | 67.74 |
| HexFormer-Hybrid | 40 | **68.67** |
| Euclidean ViT | 50 | 67.88 |
| HexFormer | 50 | 70.54 |
| HexFormer-Hybrid | 50 | **70.92** |
| Euclidean ViT | 60 | 69.54 |
| HexFormer | 60 | 72.58 |
| HexFormer-Hybrid | 60 | **72.84** |
| Euclidean ViT | 70 | 70.2 |
| HexFormer | 70 | 72.51 |
| HexFormer-Hybrid | 70 | **73.98** |

## A.4 RUNTIME, MEMORY, AND LIMITATIONS

Hyperbolic networks are generally more computationally demanding and can exhibit numerical instability, which affects both training runtime and memory usage (Mettes et al., 2024). The Exponential Map Aggregation (ExpAgg) introduces less than 5% additional computational overhead compared to centroid aggregation while improving numerical stability, as shown in Table 12. The Riemannian AdamW optimizer (Bdeir et al., 2024) further enhances training efficiency compared to standard Riemannian Adam used in previous hyperbolic ViT models as reported in their paper.

Memory usage for HexFormer and Hybrid-HexFormer is higher than for Euclidean ViT, as reported in Table 14, but remains manageable on modern GPUs. Training runtime is substantially longer for hyperbolic models. However, convergence can be achieved in fewer epochs. In the main experi-

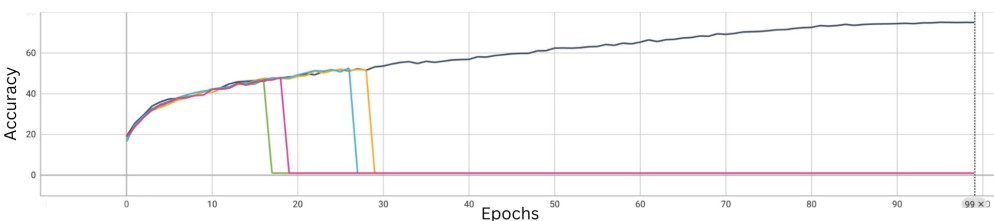

(a) HexFormer-Tiny with centroid using AdamW on CIFAR-100 for 5 different seeds

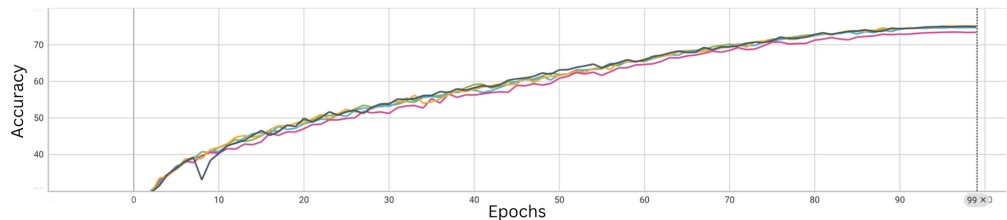

(b) HexFormer-Tiny with ExpAgg using AdamW on CIFAR-100 for 5 different seeds

Figure 10: Comparison of the same runs on AdamW on 5 different seeds between HexFormer with centroid and HexFormer with ExpAgg.

Table 12: Average runtime per iteration (ms) and iterations per second for different Tiny-ViT models on CIFAR-100.

| Model | Avg. Iteration Time | Iterations per Second |
|---|---|---|
| Euclidean ViT | 46.16 ms | 21.67 |
| HexFormer (Centroid) | 188.12 ms | 5.32 |
| HexFormer (ExpAgg) | 196.64 ms | 5.09 |
| HexFormer-Hybrid (Centroid) | 190.52 ms | 5.25 |
| HexFormer-Hybrid (ExpAgg) | 197.01 ms | 5.07 |

ments we ran everything for 100 epochs but for instance, Table 13 and results in Table 11 indicate that HexFormer without warmup reaches higher accuracy after approximately 60 epochs compared to Euclidean ViT trained for 100 epochs without warmup Table 3. Inference runtime, by contrast, scales more gracefully, approximately doubling compared to Euclidean models, and centroid variants are omitted from the tables when their performance is nearly identical to ExpAgg, indicating negligible differences in practical use.

Table 13: Inference runtime for Euclidean and Hyperbolic models on CIFAR-100 and Tiny-ImageNet.

| Dataset | Model | Total Inference Time (s) | Time per Image (ms) |
|---|---|---|---|
| CIFAR-100 | Euclidean ViT | 1.3296 s | 0.1329 ms |
| | HexFormer | 3.0991 s | 0.3099 ms |
| | HexFormer-Hybrid | 3.0614 s | 0.3061 ms |
| Tiny-ImageNet | Euclidean ViT | 4.0127 s | 0.8025 ms |
| | HexFormer | 8.2597 s | 1.6519 ms |
| | HexFormer-Hybrid | 8.1895 s | 1.6379 ms |

Forward FLOPs and the trainable parameter counts for the Tiny-ViT models are shown in Table 15. These were computed using the THOP library, which provides automated estimates of the number of floating-point operations for a given PyTorch model. THOP primarily accounts for standard linear,

Table 14: Memory usage (in GB) for Tiny-ViT models on CIFAR-100. Values show allocated and reserved GPU memory.

| Model | Allocated Memory | Reserved Memory |
|---|---|---|
| Euclidean ViT | 1.11 GB | 1.22 GB |
| HexFormer | 2.86 GB | 3.08 GB |
| HexFormer-Hybrid | 2.93 GB | 3.13 GB |

Table 15: Comparison of forward FLOPs and number of parameters for different Tiny-ViT models on CIFAR-100.

| Model | Trainable Params | Forward FLOPs |
|---|---|---|
| Euclidean ViT | 2.70 M | 0.351 GFLOPs |
| Euclidean HexFormer | 2.71 M | 0.352 GFLOPs |
| Hybrid HexFormer | 2.69 M | 0.352 GFLOPs |

convolutional, and activation operations, but it does not automatically include more complex or custom operations, such as those used in hyperbolic layers (e.g., exponential maps, logarithmic maps, and Riemannian operations). As a result, the absolute FLOPs values reported here may underestimate the true computational cost for hyperbolic models.

Overall, hyperbolic transformers are more resource-intensive, and this computational overhead is a known challenge in the field. Ongoing research is actively exploring methods to improve the efficiency of hyperbolic architectures while retaining their advantages in numerical stability and hierarchical representation.

### A.5 USE OF LARGE LANGUAGE MODELS IN THIS PAPER

Large Language Models (LLMs) were used in several supportive capacities during the preparation of this paper. Specifically, LLMs assisted in rephrasing and polishing text to improve clarity and readability, and helped structure tables and templates in a consistent format. In addition, LLMs were used for retrieval and discovery of related work to ensure that relevant literature was not overlooked. All technical content, analyses, experiments, and conclusions presented in this paper were independently generated by the authors.

