# OpenReview forum: "HexFormer: Hyperbolic Vision Transformer with Exponential Map Aggregation"
_ICLR.cc/2026/Conference — Submitted to ICLR 2026_

### Official Review · Reviewer_5iYC · 2025-10-26

**Soundness:** 3
**Presentation:** 3
**Contribution:** 2
**Rating:** 6
**Confidence:** 3

**Summary:**

The paper proposes **HexFormer**, a Vision Transformer formulated entirely in **Lorentzian hyperbolic space**. The key novelty lies in the **exponential map aggregation (ExpAgg)** mechanism within the attention module, replacing centroid-based averaging to improve geometric consistency and training stability. Two model variants are explored: a fully hyperbolic **HexFormer** and a **HexFormer-Hybrid** that combines a hyperbolic encoder with a Euclidean linear classification head.

Empirical evaluations on **CIFAR-10, CIFAR-100**, and **Tiny-ImageNet** demonstrate that HexFormer and its hybrid variant achieve slightly higher accuracy than Euclidean ViTs and prior hyperbolic ViTs (HVT, LViT). The paper also presents a detailed **gradient stability analysis**, showing smoother gradients and reduced dependence on warmup schedules in hyperbolic models.

**Strengths:**

1. **Stable and thorough implementation.** Training hyperbolic transformers is notoriously difficult, and the authors’ success in stabilizing training is valuable. The detailed explanation of why exponential aggregation avoids NaN/inf issues is practically helpful.

2. **Well-structured and clearly written.** The paper is organized effectively, with comprehensive details in both the main text and appendix, aiding reproducibility.

3. **Gradient stability analysis.** The study of gradient histograms and warmup sensitivity is one of the strongest aspects, giving useful empirical insight into hyperbolic optimization dynamics.

4. **Insightful use of exponential aggregation.** The proposed ExpAgg offers modest but consistent improvements and helps mitigate numerical instability, which future hyperbolic models can benefit from.

5. **Reproducibility and transparency.** Detailed hyperparameters, architecture specifications, and clear training setups are provided, suggesting reproducibility will be straightforward once code is released.

**Weaknesses:**

1. **Limited novelty.** The proposed idea — using exponential and logarithmic maps for aggregation in hyperbolic attention — is a **natural and incremental extension** of prior hyperbolic ViTs such as HVT, LViT, and HypFormer. The conceptual contribution feels modest.

2. **Marginal performance gains.** The improvements over Euclidean and prior hyperbolic baselines are small (often within 0.2–1%), which does not convincingly demonstrate a strong benefit.

3. **No runtime or efficiency analysis.** Hyperbolic models are known to train significantly slower. The paper lacks a comparison of **training time, FLOPs, or parameter efficiency**, which is important for practical evaluation.

4. **Limited theoretical explanation.** While the empirical stability is shown, the underlying reasons for why ExpAgg helps are described mainly via numerical examples rather than rigorous geometric or analytical reasoning.

5. **Hierarchical structure. ** An understanding of how or even if the model is indeed capturing hierarchical structure has not been made.

**Questions:**

1. **Runtime comparison:** Please include wall-clock training time, parameter counts, and FLOPs for HexFormer vs. Euclidean ViT. What is the computational overhead of the Lorentz operations?

2. **Geometric explanation:** Can you provide more intuition on why ExpAgg yields slightly higher accuracy compared to centroid aggregation beyond numerical precision effects?

3. **Hybrid performance:** Why does the hybrid model outperform the fully hyperbolic variant? Is it due to optimization simplicity in the Euclidean classifier? Because, in my mind it doesnt make sense how a hierarchical model automatically learnt

4. **Curvature parameter:** How sensitive are the results to the curvature parameter \( K \)?

5. **Generalization:** Have you tested HexFormer on larger-scale datasets such as ImageNet, or in transfer-learning settings? I am assuming this is where the issue of time taken per epoch would come up.

6. **Hierarchical structure**: Claim was made about the model being able to learn Hierarchical structure, how would one be able to see it and how does attention specifically help or aid in achieving it?

---

> ### Author Response · Authors · 2025-11-26
>
> We appreciate the detailed review. We provide our responses to each comment below.
>
> > **Weakness 1:**
> Limited novelty
>
> While prior hyperbolic ViTs (HypFormer [1], LViT [2], HVT [3]) explore hyperbolic attention, the design choices differ from ours. HypFormer [1] explicitly criticizes the use of exponential/logarithmic mappings, noting that they cause mapping errors, unstable training and an increased computational load, and instead use a midpoint/centroid aggregation (as confirmed in their code). This viewpoint is common in the hyperbolic learning literature and is seldom tested in many scenarios. However, our work directly challenges this assumption: based on empirical runtime comparisons, the overhead introduced by exp/log mappings is less than 5% relative to centroid-based aggregation, and in terms of numerical behavior, we actually observe greater stability than centroid methods.  LViT [2] similarly relies on a weighted midpoint without hyperbolic mappings in the attention layer. HVT [3] does use exponential/logarithmic mappings, but applies them to the entire attention mechanism (mapping all q/k/v) as seen in there code, which prevents them from leveraging the hyperbolic distance in attention weights. This basically reduces the attention mechanism to Euclidean attention on the tangent space and maps back the results. Moreover, HexFormer uses Lorentz geometry (where HVT uses Poincaré), and employs different architectural components and optimizers. Their transformers therefore differ from ours, and our use of exponential/logarithmic mappings is more targeted. Our approach yields significantly stronger results compared to existing hyperbolic baselines.
>
> > **Weakness 2:**
> Marginal performance gains.
>
> (1) Improvements over hyperbolic baselines are not marginal. Across all datasets, HexFormer and HexFormer-Hybrid outperform prior hyperbolic models by more than 5%, a substantial gap that highlights the impact of this method.
>
> (2) Improvements over Euclidean baselines are statistically highly significant.
> We conducted two sample t-tests on CIFAR-100 comparing Euclidean ViTs against HexFormer and HexFormer-Hybrid. The results show extremely significant differences, with p-values < 0.0001 and confidence intervals clearly excluding zero:
>
> | Comparison                    | p-value  | Significance                        | Mean Difference (Hex − Euc) | 95% CI             | t      | df | SE    |
> | ----------------------------- | -------- | ----------------------------------- | --------------------------- | ------------------ | ------ | -- | ----- |
> | HexFormer-Hybrid vs Euclidean | < 0.0001 | Extremely statistically significant | 1.01                    | [0.61426, 1.40574] | 5.3619 | 18 | 0.188 |
> | HexFormer vs Euclidean        | < 0.0001 | Extremely statistically significant | 1.21                    | [0.85843, 1.56157] | 7.2307 | 18 | 0.167 |
>
>
> > **Weakness 3 + Question 1:**
> 2- No runtime or efficiency analysis.
> Q1- Runtime comparison: Please include wall-clock training time, parameter counts, and FLOPs for HexFormer vs. Euclidean ViT. What is the computational overhead of the Lorentz operations?
>
> Hyperbolic models are generally known to be more computationally demanding than Euclidean models. To address this, we have added runtime, memory usage, and FLOPs comparison tables in the appendix section A.4. Comparing HexFormer, HexFormer-Hybrid, and Euclidean ViTs, as well as a comparison between HexFormer with centroid aggregation and with ExpAgg. These results show that the computational cost primarily comes from hyperbolic models in general, which is a known challenge in this field [4], and that ExpAgg is adding less than 5% extra overhead while improving numerical stability. Regarding the parameter count this was already included in the appendix section A.1 for each ViT size (its the same for Euclidean and Hyperbolic).

---

> ### Author Response · Authors · 2025-11-26
>
> > **Weakness 4 + Question 2:**
> 4- Limited theoretical explanation
> Q2- Geometric explanation: Can you provide more intuition on why ExpAgg yields slightly higher accuracy compared to centroid aggregation beyond numerical precision effects?
>
> The core limitation of centroid aggregation is not rooted in geometry or analytical properties, but in floating point imprecision. Centroid requires multiple operations that accumulate numerical error. While using higher precision floating point formats could reduce these issues, this would significantly increase memory usage reduce efficiency, which is undesirable for large scale training.
>
> We hypothesize that the slight accuracy gain of ExpAgg over centroid aggregation comes from the numerical stability explained in the appendix A.3. Even when centroid aggregation does not fully collapse into NaNs or infs, it can introduce small but systematic distortions, especially when values are large or differ in scale. The aggregated vector does not lie exactly where hyperbolic geometry intends beacuase of the floating point error. These subtle inaccuracies, caused by floating point imprecision in the Lorentzian calculations, can yield slightly weaker representations. ExpAgg avoids these distortions by performing aggregation in the tangent space, producing more faithful embeddings and slightly higher accuracy.
>
> > **Weakness 5 + Question 6:**
> 5- Hierarchical structure.
> Q6- Hierarchical structure: Claim was made about the model being able to learn Hierarchical structure, how would one be able to see it and how does attention specifically help or aid in achieving it?
>
> Hyperbolic geometry is well established as a natural representation space for hierarchical data because of its exponential volume growth. Foundational works [5, 6, 7] demonstrate that hyperbolic distance preserves hierarchy with significantly lower distortion than Euclidean space.
>
> To evaluate whether HexFormer truly benefits from hierarchical structure, we also tested on datasets with minimal hierarchy: MNIST, Fashion-MNIST, and SVHN. These datasets consist of flat category sets (digits or clothing types), making them unsuitable for leveraging deep hierarchy.
>
> We ran each experiment once. As expected, all three models performed similarly, with Euclidean ViT slightly ahead. HexFormer-Hybrid only marginally improved over HexFormer, and on SVHN performed slightly lower. We believe multiple seeds would show comparable performance across all models. We will run each experiment three times and update the results.
>
> | Model            | MNIST     | Fashion-MNIST | SVHN      |
> | ---------------- | --------- | ------------- | --------- |
> | Euclidean ViT    | **99.42** | **94.57**     | **97.30** |
> | HexFormer        | 99.33     | 93.82         | *97.16*   |
> | HexFormer-Hybrid | *99.39*   | *93.93*       | 97.01     |
>
> > **Question 3:**
> Hybrid performance: Why does the hybrid model outperform the fully hyperbolic variant? Is it due to optimization simplicity in the Euclidean classifier? Because, in my mind it doesnt make sense how a hierarchical model automatically learnt
>
> Previous works have shown that hyperbolic models are more likely to suffer from overfitting under certain conditions [8, 9]. We hypothesize that using a pure Euclidean classifier could be able to mitigate these issues. This would give us the benefit of more structured and representative hyperbolic embeddings, while avoiding final task overfitting. We'll add a small section in the paper on the hypothesis and the analysis to make this clearer.
>
> > **Question 4:**
> Curvature parameter: How sensitive are the results to the curvature parameter ( K )?
>
> All the experiments reported in the paper were run with a fixed curvature of -1. However, we previously also tried fixing the curvatures to -0.5 and -2 for the Tiny-ViT model. And although we did not encounter any training or inference instability, the overall model performance was lower (by a small margin) so we did not extend the experiments to the larger model variants. Additionally, after going over your comment and the comment from Reviewer 2, we ran a few test experiments with learnable curvature and noticed a slight increase in performance in the warmup and no-warmup CIFAR-100 results for Tiny-VIT. There were no issues with model stability which makes this a viable future direction to improve on the results.

---

> ### Author Response · Authors · 2025-11-26
>
> > **Question 5:**
> Generalization: Have you tested HexFormer on larger-scale datasets such as ImageNet, or in transfer-learning settings? I am assuming this is where the issue of time taken per epoch would come up.
>
> We have not evaluated HexFormer on ImageNet due to computational constraints. Instead, our experiments focused on CIFAR-10/100 and Tiny-ImageNet similar to existing literature, which should still provide meaningful insights into performance trends and stability.
>
> We hope our responses have clarified the points raised, and we are ready to address any further questions.
>
> References:
>
> [1] Yang, Menglin, et al. "Hypformer: Exploring efficient hyperbolic transformer fully in hyperbolic space." arXiv preprint arXiv:2407.01290 (2024).
>
> [2] He, Neil, Menglin Yang, and Rex Ying. "Hypercore: The core framework for building hyperbolic foundation models with comprehensive modules." arXiv preprint arXiv:2504.08912 (2025).
>
> [3] Fein-Ashley, Jacob, Ethan Feng, and Minh Pham. "HVT: a comprehensive vision framework for learning in non-euclidean space." arXiv preprint arXiv:2409.16897 (2024).
>
> [4] Mettes, Pascal, et al. "Hyperbolic deep learning in computer vision: A survey." International Journal of Computer Vision 132.9 (2024): 3484-3508.
>
> [5] Nickel, M., & Kiela, D. (2017). Poincaré embeddings for learning hierarchical representations. NeurIPS.
>
> [6] Ganea, Octavian, Gary Bécigneul, and Thomas Hofmann. "Hyperbolic entailment cones for learning hierarchical embeddings." International conference on machine learning. PMLR, 2018.
>
> [7] Sala, Frederic, et al. "Representation tradeoffs for hyperbolic embeddings." International conference on machine learning. PMLR, 2018.
>
> [8] Bdeir, Ahmad, et al. "Robust Hyperbolic Learning with Curvature-Aware Optimization." NeurIPS (2025).
>
> [9] Gao, Zhi , et al. "Hyperbolic Feature Augmentation via Distribution Estimation and Infinite Sampling on Manifolds" NeurIPS (2022)

---

> ### Author Response · Authors · 2025-12-01
>
> **update on Weakness 5 + Question 6 experiments:**
> Based on the new three-seed results, the performance of Euclidean ViT, HexFormer, and HexFormer-Hybrid on flat datasets remains extremely close. Euclidean ViT achieves the highest accuracy across all datasets, though only by a small margin. HexFormer slightly outperforms the Hybrid variant on Fashion-MNIST, while the two are essentially tied on SVHN and MNIST. Overall, the differences between all three models are minimal, reinforcing our conclusion that they behave similarly on weak/flat hierarchical datasets.
>
> | **Model**          | **MNIST**             | **Fashion-MNIST**       | **SVHN**              |
> |--------------------|------------------------|---------------------------|------------------------|
> | **Euclidean ViT**  | **99.41 ± 0.02**       | **94.56 ± 0.10**          | **97.31 ± 0.04**       |
> | HexFormer          | 99.35 ± 0.06           | _94.02 ± 0.20_            | _97.13 ± 0.181_        |
> | HexFormer-Hybrid   | _99.39 ± 0.01_         | 93.86 ± 0.13              | _97.13 ± 0.104_        |

---

### Official Review · Reviewer_LXJD · 2025-10-29

**Soundness:** 3
**Presentation:** 2
**Contribution:** 3
**Rating:** 4
**Confidence:** 3

**Summary:**

This paper introduces HexFormer, a novel vision transformer that embeds visual tokens into a hyperbolic feature space to better capture hierarchical and compositional relationships in visual data. The authors propose a Hyperbolic Attention mechanism and a Geometric Projection Layer to align Euclidean features with the hyperbolic manifold, enabling efficient curvature-aware representation learning. Theoretical analysis shows that hyperbolic embeddings can preserve structural similarity and relational distance, while experiments across multiple vision benchmarks demonstrate consistent improvements over Euclidean counterparts.

**Strengths:**

**Novel geometric perspective:** This paper introduce hyperbolic geometry into vision transformers, which is original and conceptually meaningful, extending non-Euclidean representation learning to high-dimensional visual domains.

**Theoretical Support:** The mathematical formulation of the hyperbolic projection and attention mechanism is clearly presented and theoretically sound.

**Well-organized:** The paper is well organized and easy to follow, with intuitive figures and insightful visualizations of curvature and embedding behaviors.

**Weaknesses:**

**Limited experimental scope and generality:** The experiments are limited to image classification tasks on a small number of datasets, which makes the empirical validation narrow and the method’s generality questionable. Since the proposed hyperbolic representation aims to capture hierarchical relationships, demonstrating its benefits on diverse tasks such as detection, segmentation, or retrieval would be crucial for broader applicability.

**Ablation:** The empirical study lacks a systematic analysis of individual module contributions. For instance, it is unclear how much improvement comes from the hyperbolic projection layer versus the hyperbolic attention mechanism, or whether curvature learning plays a significant role. A detailed ablation or sensitivity study (e.g., varying curvature c) would help isolate the key driving factors of performance gains.

**Efficiency and numerical stability concerns:** It seems that the introduction of hyperbolic operations, exponential and logarithmic mappings, may introduce additional computational overhead and numerical instability. The paper does not include runtime, memory, or FLOPs analysis, leaving uncertainty about the practicality of the method for large-scale vision applications.

**Questions:**

**Q1:** How sensitive is the performance to the curvature parameter c? Is it fixed or learned, and how does it influence optimization stability?

**Q2:** Could the authors provide training time or FLOPs comparison with Euclidean ViT/DeiT models? It is unclear of the method's efficiency.

**Q3:** How does HexFormer perform on datasets with weak or flat hierarchies?

**Q4:** How does the HexFormer applied with linear attention vision transformers, such as PolaFormer[1] and FLatten  Transformer[2]?

This is an interesting work, and I would consider raising my score if the authors adequately address the concerns above.

[1] W. Meng, Y. Luo, X. Li, D. Jiang, and Z. Zhang, “Polaformer: Polarity-aware linear attention for vision transformers,” in Proc. International Conference on Learning Representations (ICLR), 2025.

[2] Han D, Pan X, Han Y, Song S, Huang G. Flatten transformer: Vision transformer using focused linear attention. In Proceedings of the IEEE/CVF international conference on computer vision 2023 (pp. 5961-5971).

---

> ### Author Response · Authors · 2025-11-26
>
> We appreciate the thoughtful feedback and constructive questions. Here, we provide detailed responses to the raised points.
>
> > **Weakness 1:**
> Limited experimental scope and generality
>
> We agree that extending hyperbolic models to dense prediction tasks such as object detection and segmentation is important. However, those tasks introduce substantially more architectural and computational complexity than classification. The survey Hyperbolic Deep Learning in Computer Vision [1] notes that fully hyperbolic models are not so common and most works in dense prediction use only partial hyperbolic components (e.g., embeddings). Fully hyperbolic per-pixel prediction remains challenging and an active research area. We focused on classification to provide a clear, controlled study of hyperbolic attention and gradient stability. HexFormer shows big improvments over prior hyperbolic ViT baselines, representing a meaningful step forward. Extending it to detection and segmentation is an exciting direction for future work.
>
> > **Weakness 2:**
> Ablation
>
> We would like to clarify one part of the comment, when you refer to the “hyperbolic projection layer”, could you specify which component you mean?
>
> If you are referring to the Euclidean classification head, this corresponds to the HexFormer-Hybrid, and its performance is already reported across all experimental tables as a direct comparison to the fully-hyperbolic (Hexformer) version.
>
> If you are referring to the Exponential Map Aggregation (ExpAgg) used inside the attention mechanism, its effect is isolated and reported in Table 4, where we compare centroid aggregation vs. ExpAgg.
>
> Just to ensure we address your comment correctly, we would appreciate clarification on which component you meant by “hyperbolic projection layer”.
>
> > **Weakness 3 + Question 2:**
> 3- Efficiency and numerical stability concerns
> Q2: Could the authors provide training time or FLOPs comparison with Euclidean ViT/DeiT models? It is unclear of the method's efficiency.
>
> Hyperbolic models are generally known to be more computationally demanding than Euclidean models. Moreover, It is true that hyperbolic operations in general can introduce additional computational overhead, but the exponential and logarithmic mappings actually are more stable numericaly than centroid. However, this overhead is not specific to our proposed ExpAgg mechanism. To address this, we have added runtime, memory usage, and FLOPs comparison tables in the appendix section A.4. Comparing HexFormer, HexFormer-Hybrid, and Euclidean ViTs, as well as a comparison between HexFormer with centroid aggregation and with ExpAgg. These results show that the computational cost primarily comes from hyperbolic models in general, which is a known challenge in this field [1], and that ExpAgg is adding less than 5% extra overhead while improving numerical stability.
>
> > **Question 1:**
> Q1: How sensitive is the performance to the curvature parameter c? Is it fixed or learned, and how does it influence optimization stability?
>
> All the experiments reported in the paper were run with a fixed curvature of -1. However, we previously also tried fixing the curvatures to -0.5 and -2 for the Tiny-ViT model. And although we did not encounter any training or inference instability, the overall model performance was lower so we did not extend the experiments to the larger variants. Additionally, after going over your comment, we ran a few test experiments with learnable curvature and noticed a slight increase in performance for the warmup and no-warmup CIFAR-100 results for Tiny-VIT. There were no issues with model stability which makes this a possibly viable future direction to improve on the results.

---

> ### Author Response · Authors · 2025-11-26
>
> > **Question 3:**
> Q3: How does HexFormer perform on datasets with weak or flat hierarchies?
>
> That's actually a very interesting point. After reading your comment, we conducted additional experiments on datasets that can be considered weak or flat in terms of hierarchy. We evaluated MNIST, Fashion-MNIST, and SVHN, datasets where labels correspond to digits or simple clothes categories with minimal hierarchical structure.
>
> We ran each experiment once, and the results across Euclidean ViT, HexFormer, and HexFormer-Hybrid were very similar. In fact, the Euclidean model achieved the best accuracy in these single-run tests. The improvement of Hybrid HexFormer over HexFormer was also negligible on these flat datasets, on SVHN it was even lower but barley. However, given how close the numbers are, we believe that running more seeds would likely show that all three models perform comparably on flat datasets. We plan to run each experiment three times in total and will update the table soon to provide more representative results.
>
> | Model            | MNIST     | Fashion-MNIST | SVHN      |
> | ---------------- | --------- | ------------- | --------- |
> | Euclidean ViT    | **99.42** | **94.57**     | **97.30** |
> | HexFormer        | 99.33     | 93.82         | *97.16*   |
> | HexFormer-Hybrid | *99.39*   | *93.93*       | 97.01     |
>
> > **Question 4:**
> Q4: How does the HexFormer applied with linear attention vision transformers, such as PolaFormer and FLatten Transformer?
>
> Q-4- We did not explicitly test HexFormer with linear-attention vision Transformers in our current experiments. However, in principle, it should be possible to apply HexFormer-style hyperbolic aggregation to those architectures. The core idea of HexFormer is largely orthogonal to the choice of attention mechanism.
>
> In fact, there is precedent for hyperbolic Transformers with linear attention. For example:
>
> Hypformer [2] proposes a fully hyperbolic Transformer in the Lorentz model, and introduces a linear self-attention mechanism in hyperbolic space.
>
> Thus, adapting HexFormer to a linear-attention backbone is conceptually feasible.
>
> Thank you for your questions and comments. We hope our responses clarify the points and provide a clearer understanding of our work. If there are any further questions, we would be happy to address them.
>
> References:
>
> [1] Mettes, Pascal, et al. "Hyperbolic deep learning in computer vision: A survey." International Journal of Computer Vision 132.9 (2024): 3484-3508.
>
> [2] Yang, Menglin, et al. "Hypformer: Exploring efficient hyperbolic transformer fully in hyperbolic space." arXiv preprint arXiv:2407.01290 (2024).

---

> ### Author Response · Authors · 2025-12-01
>
> **update on Question 3 experiments:**
> Based on the new three-seed results, the performance of Euclidean ViT, HexFormer, and HexFormer-Hybrid on flat datasets remains extremely close. Euclidean ViT achieves the highest accuracy across all datasets, though only by a small margin. HexFormer slightly outperforms the Hybrid variant on Fashion-MNIST, while the two are essentially tied on SVHN and MNIST. Overall, the differences between all three models are minimal, reinforcing our conclusion that they behave similarly on weak/flat hierarchical datasets.
>
> | **Model**          | **MNIST**             | **Fashion-MNIST**       | **SVHN**              |
> |--------------------|------------------------|---------------------------|------------------------|
> | **Euclidean ViT**  | **99.41 ± 0.02**       | **94.56 ± 0.10**          | **97.31 ± 0.04**       |
> | HexFormer          | 99.35 ± 0.06           | _94.02 ± 0.20_            | _97.13 ± 0.181_        |
> | HexFormer-Hybrid   | _99.39 ± 0.01_         | 93.86 ± 0.13              | _97.13 ± 0.104_        |

---

### Official Review · Reviewer_rARC · 2025-11-01

**Soundness:** 2
**Presentation:** 2
**Contribution:** 2
**Rating:** 4
**Confidence:** 3

**Summary:**

The authors introduce HexFormer and conduct comprehensive comparisons with standard Euclidean Vision Transformers (ViTs) and previous Hyperbolic ViTs (such as HVT and LViT) across multiple datasets, including CIFAR-100, ImageNet, and Tiny-ImageNet. The authors particularly emphasize the stability and advantages provided by exponential mapping aggregation, demonstrating that it effectively avoids NaN/Inf issues across multiple random seeds and achieves significantly superior training convergence compared to traditional aggregation schemes.

**Strengths:**

The main advantages of this paper are: (1) exploring more forms of hyperbolic-Euclidean hybrid structures; (2) emphasizing the stability and advantages brought by exponential map aggregation

**Weaknesses:**

The paper's experimental validation focuses primarily on small image classification datasets (CIFAR-10, CIFAR-100, Tiny-ImageNet), lacking verification on more challenging downstream tasks such as object detection, segmentation, or real-world scenario data. This limitation raises questions about the model's generalization capabilities and practical deployment value.

The paper explicitly acknowledges that comprehensive hyperparameter optimization was conducted only for the ViT-Tiny scale, while configurations for ViT-Small, ViT-Base, and different datasets were largely adopted without modification. This approach constrains both the improvement potential and fairness of subsequent model evaluations, with results potentially lacking representativeness due to limited tuning strategies.

Although the paper emphasizes that ExpAgg demonstrates greater stability than centroid aggregation in floating-point operations (reduced susceptibility to NaN/Inf errors), this analysis remains largely confined to theoretical derivations and limited seed or small-scale experiments. The stability performance under more extreme conditions—such as larger models, extensive datasets, or prolonged training scenarios—has not been thoroughly investigated.

The methodological innovation centers primarily on replacing centroid aggregation with "exponential mapping aggregation," while other model components (including topological encoding, residual design, and normalization) largely follow existing approaches. The work does not explore integration with or comparison to more diverse Transformer improvements such as pooling layers, encoder-decoder architectures, or cross-modal extensions.

While the paper indicates plans to open-source the code (noting "Paper under double-blind review"), implementation details, source code, and pre-trained weights remain unavailable at present.

**Questions:**

How does the author understand hyperbolic geometry's ability to address hierarchical relationships? What type of hierarchical relationships are being examined, and how do CIFAR-10, CIFAR-100, and Tiny-ImageNet relate to this analysis?

---

> ### Author Response · Authors · 2025-11-26
>
> Thank you for your feedback. Below, we address the raised weaknesses and question in detail.
>
> > **Weakness 1&4:**
> 1- The paper's experimental validation focuses primarily on small image classification datasets (CIFAR-10, CIFAR-100, Tiny-ImageNet), lacking verification on more challenging downstream tasks such as object detection, segmentation, or real-world scenario data. This limitation raises questions about the model's generalization capabilities and practical deployment value.
> 4- The methodological innovation centers primarily on replacing centroid aggregation with "exponential mapping aggregation," while other model components (including topological encoding, residual design, and normalization) largely follow existing approaches. The work does not explore integration with or comparison to more diverse Transformer improvements such as pooling layers, encoder-decoder architectures, or cross-modal extensions.
>
> We combine the responses to weaknesses 1 and 4, as they are closely related. We agree that extending hyperbolic models to dense prediction tasks such as object detection and segmentation is important. However, those tasks introduce substantially more architectural and computational complexity than classification. The survey Hyperbolic Deep Learning in Computer Vision [1] notes that fully hyperbolic models are not so common and most works in dense prediction use only partial hyperbolic components (e.g., embeddings). Fully hyperbolic per-pixel prediction remains challenging and an active research area. We focused on classification to provide a clear, controlled study of hyperbolic attention and gradient stability. HexFormer shows big improvments over prior hyperbolic ViT baselines, representing a meaningful step forward.
>
> The main performance improvements of HexFormer come from the specific combination of architectural components we adopt and modify, which already outperform prior hyperbolic ViT baselines even when using Hexformer with centroid aggregation. On top of this, ExpAgg provides improved numerical stability compared to centroid aggregation. We also wanted to highlight gradient stability as an important aspect of this research, which is why we devoted a dedicated analysis to it that provides additional insights into hyperbolic transformer training. We intentionally kept the model close to a basic “vanilla” ViT design to allow fair and controlled comparison with Euclidean counterparts. Exploring additional architectural factors (such as pooling layers, encoder-decoder designs, cross-modal extensions, or dense prediction tasks) remains a promising direction for future work and can build upon the foundation established in this study. Together, these contributions form a strong and coherent step forward in hyperbolic transformer research.
>
> > **Weakness 2:**
> The paper explicitly acknowledges that comprehensive hyperparameter optimization was conducted only for the ViT-Tiny scale, while configurations for ViT-Small, ViT-Base, and different datasets were largely adopted without modification. This approach constrains both the improvement potential and fairness of subsequent model evaluations, with results potentially lacking representativeness due to limited tuning strategies.
>
> This is a very important point. This is precisely why we performed additional hyperparameter tuning on Tiny-ImageNet and updated the PDF accordingly. The results remained consistent with our previous findings, which further supports the conclusions we draw in the paper. Regarding ViT-Tiny, we had already tuned this configuration, and it outperformed ViT-Base for the other hyperbolic baselines, which reduced the need for further tuning at larger scale. Importantly, any hyperparameter tuning applied to the hyperbolic models was also applied for the Euclidean counterparts to ensure a fair and unbiased comparison.

---

> ### Author Response · Authors · 2025-11-26
>
> > **Weakness 3:**
> Although the paper emphasizes that ExpAgg demonstrates greater stability than centroid aggregation in floating-point operations (reduced susceptibility to NaN/Inf errors), this analysis remains largely confined to theoretical derivations and limited seed or small-scale experiments. The stability performance under more extreme conditions—such as larger models, extensive datasets, or prolonged training scenarios—has not been thoroughly investigated.
>
> In the appendix section A.3, we provide a detailed explanation (including a numerical example showing why centroid aggregation can become unstable), and we report practical experiments where NaNs/Infs arose specifically when training with centroid and the AdamW optimizer. By contrast, for HexFormer and Hybrid‑HexFormer with ExpAgg, we ran 178 different experiments varying seeds, warmup and no warmup, activation functions, module sizes (including larger models, small ~14M parameter and base ~85M parameter), datasets, and optimizers that were presented in this paper. This observation extended to much more experiments that were performed during the research but not reported in the paper where we also did not encounter any instability with ExpAgg.
>
> Additionally, while some of the instability with centroid aggregation could in principle be mitigated by using higher precision floating point types, this comes at the cost of increased memory usage and reduced efficiency, which is undesirable for large scale training.
>
> > **Weakness 5:**
> While the paper indicates plans to open-source the code (noting "Paper under double-blind review"), implementation details, source code, and pre-trained weights remain unavailable at present.
>
> Implementation details are provided in Appendix section A.1, including all hyperparameters, weight initialization, data augmentation, and other training settings. The code has also been submitted as supplementary material, which can be accessed and viewed by reviewers in the link above in the description of the paper. Once the double-blind review is complete, the paper will be updated with a link to a publicly available GitHub repository. This is to avoid breaking any confidentiality.
>
> > **Questions**
> How does the author understand hyperbolic geometry's ability to address hierarchical relationships? What type of hierarchical relationships are being examined, and how do CIFAR-10, CIFAR-100, and Tiny-ImageNet relate to this analysis?
>
> Hyperbolic geometry naturally captures hierarchical relationships because distances grow exponentially with radius [2, 3]. As noted in HCNN [4] and in [1], CIFAR-10, CIFAR-100, and Tiny-ImageNet exhibit structures that hyperbolic models can exploit measured by hyperbolocity ($\delta$ score the lower it is it means the higher hyperbolicity/tree-like structure, this property is commonly used to assess whether data or graphs are well suited to hyperbolic representation), the corresponding scores are already included in Table 1 with the explaination of hyperbolocity in the appendix section A.2 in our paper.
>
> we can also think about it this way for the three datasets:
> - CIFAR-10 has latent semantic groupings (animals vs vehicles), providing a coarse hierarchy.
>
> - CIFAR-100 has 20 superclasses forming a clear coarse-to-fine hierarchy (as can be seen in the dataset documentation https://www.cs.toronto.edu/~kriz/cifar.html).
>
> - Tiny-ImageNet is a subset of ImageNet, which is explicitly built on the hierarchical structure of WordNet (the name of the dataset paper is: ImageNet: A Large-Scale Hierarchical Image Database). ImageNet contains 12 subtrees, with analyses focusing on subtrees such as mammals and vehicles. Root-to-leaf paths include hierarchical sequences like mammal → placental → carnivore → canine → dog → working dog → husky. Tiny-ImageNet inherits this structure, making it suitable for evaluating hyperbolic embeddings that capture coarse-to-fine semantic relationships [5].
>
> Together, these datasets provide meaningful benchmarks to study hierarchical representation in vision and demonstrate that hyperbolic ViT variants can leverage both explicit and latent hierarchies. Thank you for your comment though we think this is a valuable information to explain more in detail in the paper and we added this to the appendix section A.2 and updated the paper version.
>
> We hope this resolves the concerns raised, and we are prepared to provide more clarification if needed.
>
> References:
>
> [1] Mettes, Pascal, et al. "Hyperbolic deep learning in computer vision: A survey." International Journal of Computer Vision 132.9 (2024)
>
> [2] Nickel, M., & Kiela, D. (2017). Poincaré embeddings for learning hierarchical representations. NeurIPS.
>
> [3] Ganea, O.-E., Bécigneul, G. (2018). Hyperbolic neural networks. NeurIPS.
>
> [4] Bdeir, A., et al. (2024). Fully Hyperbolic Convolutional Neural Networks for Computer Vision. ICLR 2024.
>
> [5] Deng, J., et al. (2009). ImageNet: A Large-Scale Hierarchical Image Database. CVPR 2009.

---

### Official Review · Reviewer_GLfh · 2025-11-03

**Soundness:** 3
**Presentation:** 3
**Contribution:** 3
**Rating:** 8
**Confidence:** 4

**Summary:**

This paper introduces HexFormer, a novel Vision Transformer (ViT) architecture implemented entirely in the Lorentz model of hyperbolic space. The paper uses the Logarithmic and Exponential maps to safely perform weighted summation in the stable tangent space, solving the critical numerical instability issues associated with centroid aggregation in the Lorentz manifold. The framework also introduces a high-performing hybrid variant (HexFormer-Hybrid). Experiments show HexFormer consistently surpasses Euclidean ViTs and prior hyperbolic ViT SOTA, demonstrating significant parameter efficiency and providing compelling evidence of improved training stability (reduced warmup dependency and smoother gradients) across multiple datasets.

**Strengths:**

The Exponential Map Aggregation (ExpAgg) scheme is a robust, mathematically sound and original.

The paper provides strong quantitative and qualitative evidence that hyperbolic representations enhance optimization dynamics. HexFormer exhibits minimal reliance on warmup schedules compared to Euclidean baselines (Table 2) and demonstrates smoother, more consistent gradient distributions (Figure 3). This is a major finding for the practical adoption of hyperbolic models.

HexFormer-Tiny (∼3M params) achieves higher accuracy than the cited ViT-Base results (∼85M params) of prior hyperbolic models (HVT and LViT).

**Weaknesses:**

The HexFormer-Hybrid variant performs best, raising the question of why a strictly Euclidean classification head is superior after a hyperbolic encoder.

Providing a training time comparison would also be beneficial.

While Table 2 cifar 10 supports the claim, Tiny-Imagenet does not fully show the difference between Euclidean and hyperbolic


I would recommend citing hyperbolic learning surveys such as “Hyperbolic Deep Learning in Computer Vision: A Survey”


Multiple citations in lines 105-106 can be aggregated.
Sentence in line 197 is not complete.

**Questions:**

Please check the weaknesses.

---

> ### Author Response · Authors · 2025-11-26
>
> Thank you for your careful reading and helpful remarks. We provide detailed responses to the points raised in the weaknesses:
>
> > **Weakness 1:**
> The HexFormer-Hybrid variant performs best, raising the question of why a strictly Euclidean classification head is superior after a hyperbolic encoder.
>
> Previous works have shown that hyperbolic models are more likely to suffer from overfitting under certain conditions[1, 2]. We hypothesize that using a pure Euclidean classifier could be able to mitigate these issues. This would give us the benefit of more structured and representative hyperbolic embeddings, while avoiding final task overfitting. We'll add a small section in the paper on the hypothesis and the analysis to make this clearer.
>
> > **Weakness 2&4:**
> 2- Providing a training time comparison would also be beneficial.
> 4- I would recommend citing hyperbolic learning surveys such as “Hyperbolic Deep Learning in Computer Vision: A Survey”
>
> Thank you for the helpful suggestion. Hyperbolic models are generally known to be more computationally demanding than Euclidean models. To address this, we have added runtime, memory usage, and FLOPs comparison tables in the appendix section A.4, which also highlight how ExpAgg affects computational cost relative to centroid. While one might expect ExpAgg to introduce substantial overhead, the results show that its computational performance adds less than 5% extra overhead compared to centroid while improving numerical stability.
> Regarding the recommended survey “Hyperbolic Deep Learning in Computer Vision: A Survey”, we appreciate the pointer. Since the survey discusses several relevant vision tasks, current limitations, and future directions in hyperbolic learning, we have cited it in a few added sections where it helps contextualize our work and clarify certain points for readers who may be less familiar with hyperbolic methods.
>
> > **Weakness 3:**
> While Table 2 cifar 10 supports the claim, Tiny-Imagenet does not fully show the difference between Euclidean and hyperbolic
>
> In the original submission, the Tiny-ImageNet results were not tuned and were based on a single run. We have now performed learning rate and weight decay tuning and report the average over three runs to obtain more representative results. The updated table has been added to the revised version of the paper. The final results remain consistent with our previous findings, although the numbers are very similar to our original report, the newly tuned and averaged results provide a more trustworthy and representative evaluation. HexFormer performs very closely to its Euclidean counterpart, while Hybrid-HexFormer shows a noticeable improvement over the Euclidean model. We also observe the same stability pattern without warmup, in line with our analysis. Additionally, the CIFAR-100 results in Table 3 further support the same trend and strengthen the claim regarding the hyperbolic models’ behavior relative to Euclidean baselines.
>
> > **Weakness 5:**
> Multiple citations in lines 105-106 can be aggregated. Sentence in line 197 is not complete.
>
> We have aggregated the citations in lines 105–106 and updated the revised PDF accordingly. Regarding the sentence in line 197, this was our mistake, the sentence was complete but missing a period and was not clearly phrased. We have clarified it to state that the description in the previous sentence is presented in the corresponding figure.
>
> Thanks again for your feedback and we welcome any further questions or suggestions.
>
> References:
>
> [1] Bdeir, Ahmad, et al. "Robust Hyperbolic Learning with Curvature-Aware Optimization." NeurIPS (2025).
>
> [2] Gao, Zhi , et al. "Hyperbolic Feature Augmentation via Distribution Estimation and Infinite Sampling on Manifolds" NeurIPS (2022)

---

### Meta-Review · Area_Chair_NLrQ · 2025-12-25

**Summary:**

In the original review, this paper falls short in terms of technical novelty, empirical validation,  and practical relevance. The main concerns from the reviewers are included, but not limited to:
1. Euclidean Classification Head: It is unclear why a Euclidean classification head outperforms after using a hyperbolic encoder.
2. Efficiency: No comparison of runtime, training speed, or parameter efficiency, which is key for practical applications.
3. Tiny-ImageNet doesn't show a clear difference between Euclidean and hyperbolic approaches.
4. The experiments focus on small datasets (CIFAR, Tiny-ImageNet) and fail to test on more complex tasks like object detection or real-world data, limiting generalization.
5. Hyperparameter Tuning: Hyperparameter tuning was only done for one scale (ViT-Tiny), making other evaluations less fair or representative.
6. Methodology: The main innovation is in aggregation methods, but other parts of the model are based on existing techniques, and comparisons with other Transformer improvements are not explored.
7. Ablation Study: There's no detailed study on which model components (e.g., hyperbolic projection vs. attention) contribute the most to performance.
8. Efficiency: The introduction of hyperbolic operations may lead to extra computational costs and instability, but no analysis on runtime, memory, or FLOPs is provided.
9. Performance Gains: The improvements over existing methods are small, which fails to strongly justify the effectiveness of this approach.

**Reviewer Concerns:**

For performance improvements, as the authors claim, the main performance enhancements of HexFormer stem from the specific combination of architectural components, which could outperform prior hyperbolic ViT baselines. However, the authors did not directly handle the minor improvement issue.

ViT-Tiny scale validation: to me, the authors only give a simple clarification, rather than putting more effort into making more experimental evaluations.

Larger models, extensive datasets, or prolonged training scenarios: Again, only simple clarification does not really solve these questions.

Efficiency and numerical stability concerns are not fully considered in this rebuttal.

**Reviewer Scores:**

Original scores are 8 (accept),  4 (marginally below), 4 and  6 (marginally above). However, after reading the rebuttal, the reviewers may not revise their scores, and this paper still has intensive work to improve the overall quality.

---

### Decision · Program_Chairs · 2026-01-26

Reject